# THE EFFECT OF DIVERSITY IN META-LEARNING

## ABSTRACT

Few-shot learning aims to learn representations that can tackle novel tasks given a small number of examples. Recent studies show that task distribution plays a vital role in the performance of the model. Conventional wisdom is that task diversity should improve the performance of meta-learning. In this work, we find evidence to the contrary; we study different task distributions on a myriad of models and datasets to evaluate the effect of task diversity on meta-learning algorithms. For this experiment, we train on multiple datasets, and with three broad classes of meta-learning models - Metric-based (i.e., Protonet, Matching Networks), Optimization-based (i.e., MAML, Reptile, and MetaOptNet), and Bayesian meta-learning models (i.e., CNAPs). Our experiments demonstrate that the effect of task diversity on all these algorithms follows a similar trend, and task diversity does not seem to offer any benefits to the learning of the model. Furthermore, we also demonstrate that even a handful of tasks, repeated over multiple batches, would be sufficient to achieve a performance similar to uniform sampling and draws into question the need for additional tasks to create better models.

## 1 INTRODUCTION

It is widely recognized that humans can learn new concepts based on very little supervision, i.e., with few examples (or "shots"), and generalize these concepts to unseen data as mentioned by Lake et al. (2011). Recent advances in deep learning, on the other hand, have primarily relied on datasets with large amounts of labeled examples, primarily due to overfitting concerns in low data regimes. Although the development of better data augmentation and regularization techniques can alleviate these concerns, many researchers now assume that future breakthroughs in low data regimes will emerge from meta-learning, or "learning to learn." Here, we study the effect of task diversity in the low data regime and its effect on various models. In this meta-learning setting, a model is trained on a handful of labeled examples at a time under the assumption that it will learn how to correctly project examples of different classes and generalize this knowledge to unseen labels at test time.

Although this setting is often used to illustrate the remaining gap between human capabilities and machine learning, we could argue that the domain of meta-learning is still nascent. The domain of task selection has remained virtually unexplored in this setting.

Conventional wisdom is that the performance of the model will improve as we train on more diverse tasks. To test this hypothesis to its limits, we define various task samplers which either limit task diversity by selecting a subset of overall tasks or improving task diversity using approaches such as Determinantal Point Processes (DPPs) proposed by Macchi (1975).

Our contributions in this work are as follows:

- We show that, against conventional wisdom, task diversity does not significantly boost performance in meta-learning. Instead, limiting task diversity and repeating the same tasks over the training phase allows the model to obtain performances similar to models trained on Uniform Sampler without any adverse effects.

- We also show that increasing task diversity using sophisticated samplers such as DPP or Online Hard Task Mining (OHTM) Samplers do not significantly boost performance. Instead, the dynamic-DPP Sampler harms the model due to the increased task diversity.

- We empirically show that repeating tasks over the training phase can perform similarly to a model trained on the Uniform Sampler, achieving similar performance with only a

fragment of data. This key finding questions the need to increase the support set pool to improve the model's performance.

## 2 RELATED WORKS

Meta-learning formulations typically rely on episodic training, wherein an algorithm adapts to a task, given its support set, to minimize the loss incurred on the query set. Meta-learning methods differ in terms of the algorithms they learn, and can be broadly classified under four prominent classes: *Metric-based*, *Model-based*, *Optimization-based* and *Bayesian-based* approaches. *Metric-based methods* such as Koch et al. (2015); Vinyals et al. (2016); Snell et al. (2017); Sung et al. (2018) operate on the core idea similar to nearest neighbors algorithm and kernel density estimation. These methods are also called non-parametric approaches. *Model-based methods* such as Santoro et al. (2016); Munkhdalai & Yu (2017) depend on a model designed specifically for fast learning, which updates its parameters rapidly with a few training steps, achieved by its internal architecture or controlled by another meta-learner model. Generic deep learning models learn through backpropagation of gradients, which are neither designed to cope with a small number of training samples nor converge within a few optimization steps. To address this, *Optimization-based methods* such as Ravi & Larochelle (2016); Finn et al. (2017); Nichol et al. (2018) were proposed, which were better suited to learn from a small number of samples. However, all the above approaches are deterministic and are not the most suited for few-shot problems that are generally ambiguous. Hence, *Bayesian-based methods* such as Yoon et al. (2018); Requeima et al. (2019) were proposed which helped address the above issue.

Although research in meta-learning models has attracted much attention recently, the effect of task diversity is virtually unexplored in the domain of meta-learning. However, task sampling and task diversity have been more extensively studied in other closely related problems such as active learning. Active learning involves selecting unlabeled data items in order to improve an existing classifier. Although most of the approaches in this domain are based on heuristics, there are few approaches to sample a batch of samples for active learning. Ravi & Larochelle (2018) proposed an approach to sample a batch of samples using a protonet as the backbone architecture. The model tries to maximize the query set, given support set and unlabeled data. Other works such as Hsu et al. (2018) proposed a framework named CACTUs, which samples tasks/examples using relatively simple task construction mechanisms such as clustering embeddings. The unsupervised representations learned via these samples lead to a good performance on various downstream human-specified tasks.

Although nascent, a few recent works aim to improve meta-learning by explicitly looking at the task structure and relationships. Among these, Yin et al. (2019) proposed an approach to handle the lack of mutual exclusiveness among different tasks through an information-theoretic regularized objective. In addition, several popular meta-learning methods Lee et al. (2019); Snell et al. (2017) improve the meta-test performance by changing the number of ways or shots of the sampled meta-training tasks, thus increasing the complexity and diversity of the tasks. Other works such as Liu et al. (2020a) proposed an approach to sample classes using class-pair-based sampling and class-based sampling. The Class-pair based Sampler selects pairs of classes that confuse the model the most. The class-based Sampler samples each class independently and does not consider the task's difficulty as a whole. Our OHTM sampler is similar to the Class-pair based Sampler. Other works such as Liu et al. (2020b) propose to augment the set of possible tasks by augmenting the pre-defined set of classes that generate the tasks with varying degrees of rotated inputs as new classes. Other works such as Setlur et al. (2020) look at the structure and diversity of tasks specifically through the lens of support set diversity, and show that, surprisingly, reducing diversity (by fixing support set) not only maintains—but in many cases, significantly improves—the performance of meta-learning. This experiment is very similar to our No Diversity Task Sampler if the size of the support set is equal to the number of classes per task. However, in this work, we extend their work on MetaOptNet, Protonet to many other models and a myriad of samplers to better understand task diversity in meta-learning. To the best of our knowledge, we are the first to study the effect of task diversity in meta-learning to this extent.

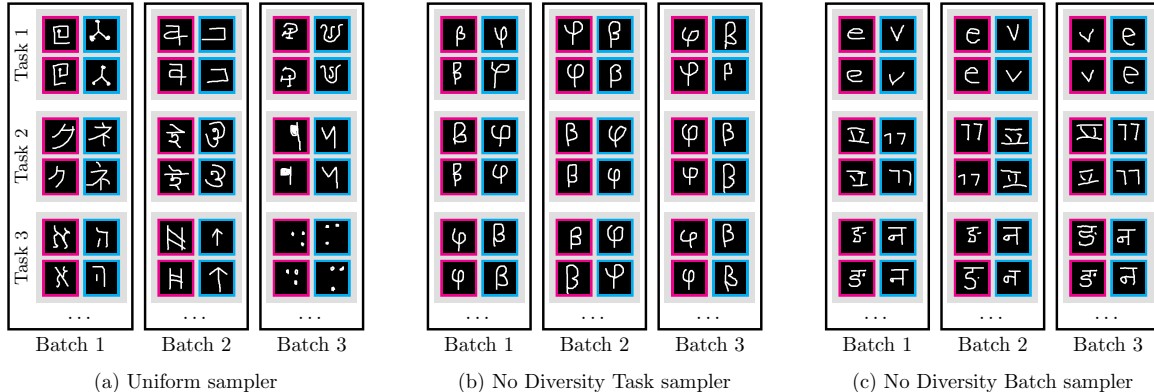

Figure 1: Illustration of (a) the Uniform Sampler, (b) the No Diversity Task Sampler, and (c) the No Diversity Batch Sampler.

## 3 BACKGROUND

Here, we review some of the fundamental ideas required to understand our few-shot learning experiments better.

### 3.1 EPISODIC FEW-SHOT LEARNING

In episodic few-shot learning, an episode is represented as a K-way, N-shot classification problem where N is the number of examples per class and K is the number of unique class labels. During training, the data in each episode is provided as a support set $S = \{(x_{1,1}, y_{1,1}), ..., (x_{N,K}, y_{N,K})\}$ where $x_{i,j} \in \mathbb{R}^D$ is the i-th instance of the j-th class, and $y_j \in \{0, 1\}^K$ is its corresponding one-hot labeling vector. Each episode aims to optimize a function $f$ that classifies new instances provided through a "query" set $Q$, containing instances of the same class as $S$. This task is difficult because $N$ is typically very small (e,g, 1 to 10). The classes change every episode. The actual test set used to evaluate a model does not contain classes seen in support sets during training. In the task-distribution view, meta-learning is a general-purpose learning algorithm that can generalize across tasks and ideally enable each new task to be learned better than the last. We can evaluate the performance of $\omega$ over a distribution of tasks $p(\mathcal{T})$. Here we loosely define a task to be a dataset and loss function $\mathcal{T} = \{\mathcal{D}, \mathcal{L}\}$. Learning how to learn thus becomes:

$$\min_{\omega} \mathbb{E}_{\tau \sim p(\tau)} \mathcal{L}(\mathcal{D}; \omega) \tag{1}$$

where $\mathcal{L}(\mathcal{D}; \omega)$ measures the performance of a model trained using $\omega$ on dataset $\mathcal{D}$ and $p(\tau)$ indicates the task distribution. In this experiment, we extend this setting such that we vary the task diversity in the train split to study the effects on test split, which remains to use uniform or random sampling for tasks.

### 3.2 DETERMINANTAL POINT PROCESSES (DPPs)

A DPP is a probability distribution over subsets of a ground set $\mathcal{Y}$, where we assume $\mathcal{Y} = \{1, 2, ..., N\}$ and $N = |\mathcal{Y}|$. An L-ensemble defines a DPP using a real, symmetric, and positive-definite matrix $\mathbf{L}$ indexed by the elements of $\mathcal{Y}$. The probability of sampling a subset $Y = A \subseteq \mathcal{Y}$ can be written as:

$$P(Y = A) \propto \det \mathbf{L}_A, \tag{2}$$

where $\mathbf{L}_A := [L_{i,j}]_{i,j \in A}$ is the restriction of $\mathbf{L}$ to the entries indexed by the elements of A. As $\mathbf{L}$ is a positive semi-definite, there exists a $d \times N$ matrix $\Psi$ such that $\mathbf{L} = \Psi^T \Psi$ where $d \leq N$. Using this principle, we define the probability of sampling as:

$$P(Y = A) \propto \det \mathbf{L}_A = Vol^2(\{\Psi_i\}_{i \in A}), \tag{3}$$

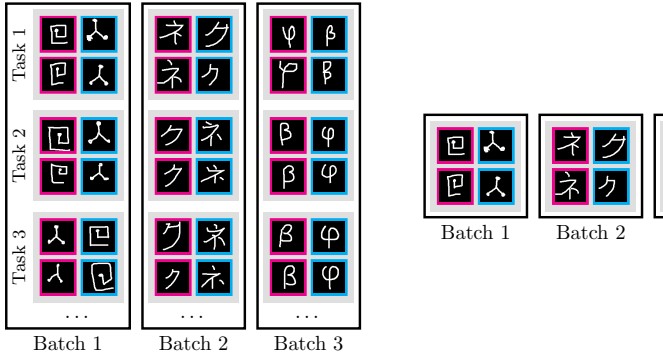

(a) No Diversity Tasks per Batch sampler  (b) Single Batch Uniform sampler

Figure 2: Illustration of (a) the No Diversity Task per Batch Sampler, and (b) the Single Batch Uniform Sampler.

where the RHS is the squared volume of the parallelepiped spanned by $\{\Psi_i\}_{i \in A}$. In Eq. 3, $\Psi_i$ is defined as the feature vector of element $i$, and each element $L_{i,j}$ in $\mathbf{L}$ is the similarity measured by dot products between elements $i$ and $j$. Hence, we can verify that a DPP places higher probabilities on diverse sets because the more orthogonal the feature vectors are, the larger the volume parallelepiped spanned by the feature vector is. In this work, these feature embeddings represent class embeddings, which are derived using either a pre-trained protonet model or the model being trained as discussed in Sec. 3.3.

In a DPP, the cardinality of a sampled subset, $|A|$, is random in general. A *k*-DPP is an extension of the DPP proposed in the work of Kuhn et al. (2003), where the cardinality of subsets are fixed as $k$ (i.e., $|A| = k$). In this work, we use *k*-DPPs as an off-the-shelf implementation to retrieve classes that represent a task used in the meta-learning step.

## 3.3 TASK SAMPLING

In this work, we experiment with eight distinct task samplers, each offering a different level of task diversity. To demonstrate the task samplers, we use a 2-way classification problem with a meta-batch size of 2 and denote each class with a unique alphabet from the Omniglot dataset.

**Uniform Sampler** This is the most widely used Sampler used in the setting of meta-learning. The Sampler gives equal probability to every task and is intuitively a random sampler. An illustration of this Sampler is shown in Figure 1.

**No Diversity Task Sampler** In this setting, we uniformly sample one set of the task at the beginning and propagate the same task across all batches and meta-batches. Note that repeating the same class over and over again does not simply repeat the same images/inputs as we episodically retrieve different images for each class. An illustration of this Sampler is shown in Figure 1.

**No Diversity Batch Sampler** In this setting, we uniformly sample one set of tasks for batch one and propagate the same tasks across all other batches. Furthermore, we shuffle these tasks to enforce that the model does not overfit. An illustration of this Sampler is shown in Figure 1.

**No Diversity Tasks per Batch Sampler** In this setting, we uniformly sample one set of tasks for a given batch and propagate the same tasks for all meta-batches. We then repeat this same principle for sampling the next batch. Furthermore, we shuffle these tasks to enforce that the model does not overfit. An illustration of this Sampler is shown in Figure 2.

**Single Batch Uniform Sampler** In this setting, we set the meta-batch size to one. This Sampler is intuitively the same as no diversity task per batch sampler, without the repetition of tasks. This

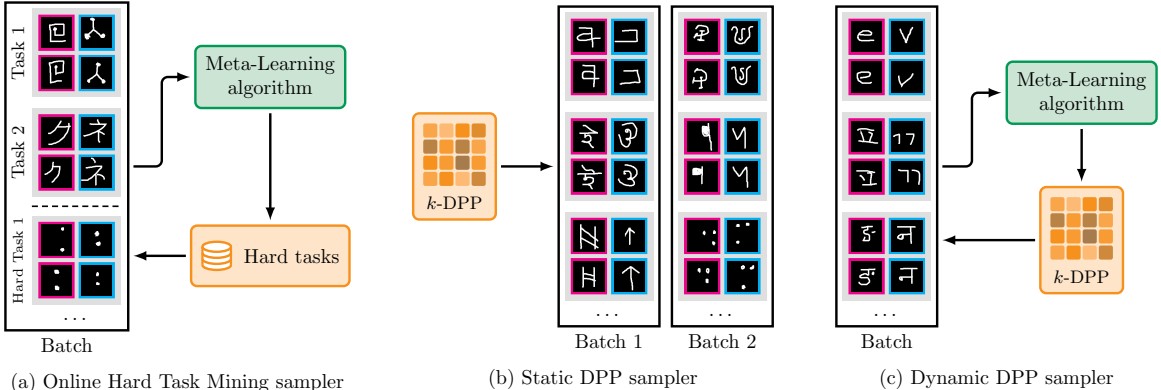

Figure 3: Illustration of (a) Online Hard Task Mining Sampler, (b) the Static DPP Sampler, and (c) the Dynamic DPP Sampler.

Sampler would be an ideal ablation study for the repetition of tasks in the meta-learning setting. An illustration of this Sampler is shown in Figure 2.

**Online Hard Task Mining Sampler**   This setting is inspired by the works of Shrivastava et al. (2016) where they proposed OHEM, which yielded significant boosts in detection performance on benchmarks like PASCAL VOC 2007 and 2012. However, to reproduce OHEM for meta-learning, we only apply the OHEM sampler for half the meta-batch size and uniform sampler for the remaining half. This approach would allow us to involve many tasks and not restrict us to only known tasks. Furthermore, to avoid OHEM in the initial stages, we sample tasks with a uniform sampler until the buffer of tasks seen by the model becomes sufficiently big, say 50 in our case. An illustration of this Sampler is shown in Figure 3.

**Static DPP Sampler**   Determinantal Point Processes (DPP) have been used for several machine learning problems such as the works of Kulesza & Taskar (2012). They have also been used in other problems such as the active learning settings in the works of Bıyık et al. (2019) and mini-batch sampling problems in the works of Zhang et al. (2019). These algorithms have also inspired other works in active learning in the batch mode setting, such as Ravi & Larochelle (2018). In this setting, we use DPP as an off-the-shelf implementation to sample tasks based on task embeddings. These task embeddings are generated using our pre-trained protonet model. The DPP instance is used to sample the most diverse tasks based on these embeddings and used for meta-learning. An illustration of this Sampler is shown in Figure 3.

**Dynamic DPP Sampler**   In this setting, we extend the previous sDPP setting such that the model in training generates the task embeddings. The Sampler is motivated by the intuition that selecting the most diverse tasks for a given model will help learn better. Furthermore, to avoid DPP in the initial stages, we sample tasks with a uniform sampler until the model becomes sufficiently trained, say 500 batches in our case. An illustration of this Sampler is shown in Figure 3.

## 4   EXPERIMENTS

The experiment aims to answer the following questions: (a) How does task diversity affect meta-learning? (b) Do sophisticated samplers such as OHEM or DPP offer any significant boost in performance? (c) Are there any rule of thumb or general good practices when it comes to sampling tasks?

To make an exhaustive study on the effect of task diversity in meta-learning, we train on four datasets: Omniglot Lake et al. (2011), *mini*Imagenet Ravi & Larochelle (2016), *tiered*ImageNet Ren et al. (2018), and Meta-Dataset Triantafillou et al. (2019). With this selection of datasets, we cover both simple datasets, such as Omniglot and *mini*ImageNet, as well as the most difficult ones, such as *tiered*ImageNet and Meta-Dataset. We train three broad classes of meta-learning models on

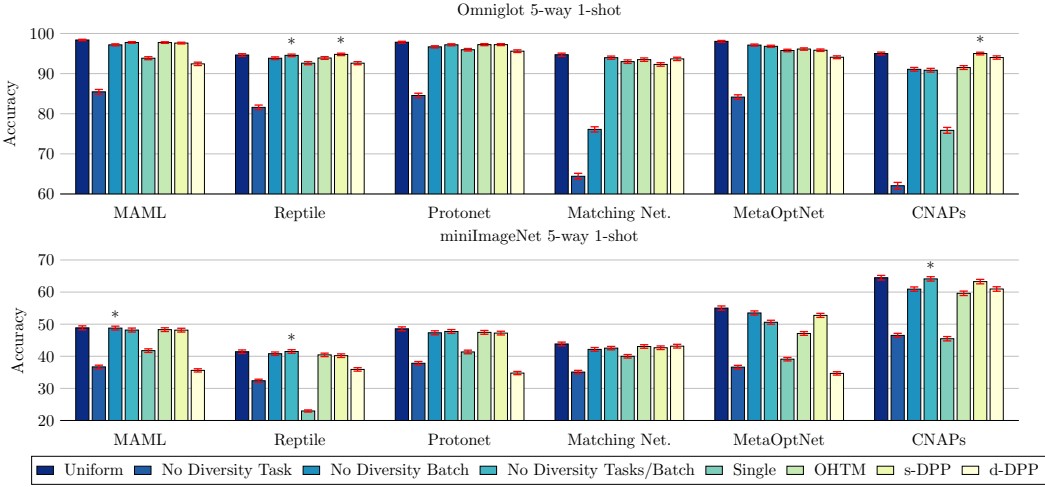

Figure 4: Average accuracy on Omniglot 5-way 1-shot & *mini*ImageNet 5-way 1-shot, with 95% confidence interval. All samplers are poorer than the Uniform Sampler and are statistically significant (with a p-value $p = 0.05$). We use the symbol $*$ to represent the instances where the results are not statistically significant and similar to the performance achieved by Uniform Sampler.

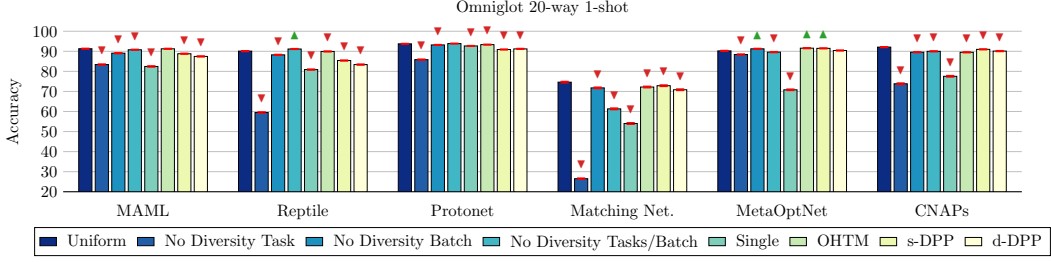

Figure 5: Average accuracy on Omniglot 20-way 1-shot, with a 95% confidence interval. We denote all samplers that are worse than the Uniform Sampler and are statistically significant (with a p-value $p = 0.05$) with ▼, and those that are significantly better than the Uniform Sampler with ▲.

these datasets - Metric-based (i.e., Protonet, Matching Networks), Optimization-based (i.e., MAML, Reptile, and MetaOptNet), and Bayesian meta-learning models (i.e., CNAPs). More details about the datasets which were used in our experiments are discussed in App. A.1. More details about the models and their hyperparameters are discussed in App. A.2. We created a common pool of 1024 randomly sampled held-out tasks to test every algorithm in our experiments to make an accurate comparison. For all experiments, we assessed the statistical significance of our results based on a paired-difference t-test, with a p-value $p = 0.05$.

## 4.1 RESULTS

In this section, we present the results of our experiments. Figure 4 presents the performance of the six models on the Omniglot and *mini*ImageNet under different task samplers in the 5-way 1-shot setting. Table 2 in the Appendix presents the same results with higher precision.

We also reproduce our experiments on the 20-way 1-shot setting on the Omniglot dataset to establish that these trends are shared across different settings. Figure 5 presents our performance of the models under this setting. Furthermore, the results on the 20-way 1-shot experiments are presented in Table 3 in the Appendix with higher precision. To further establish our findings, we also present our result on notoriously harder datasets such as *tiered*ImageNet and Meta-Dataset. Figure 6 presents the performance of the models on the *tiered*ImageNet. Again, Table 2 in the Appendix presents the same results with higher precision.

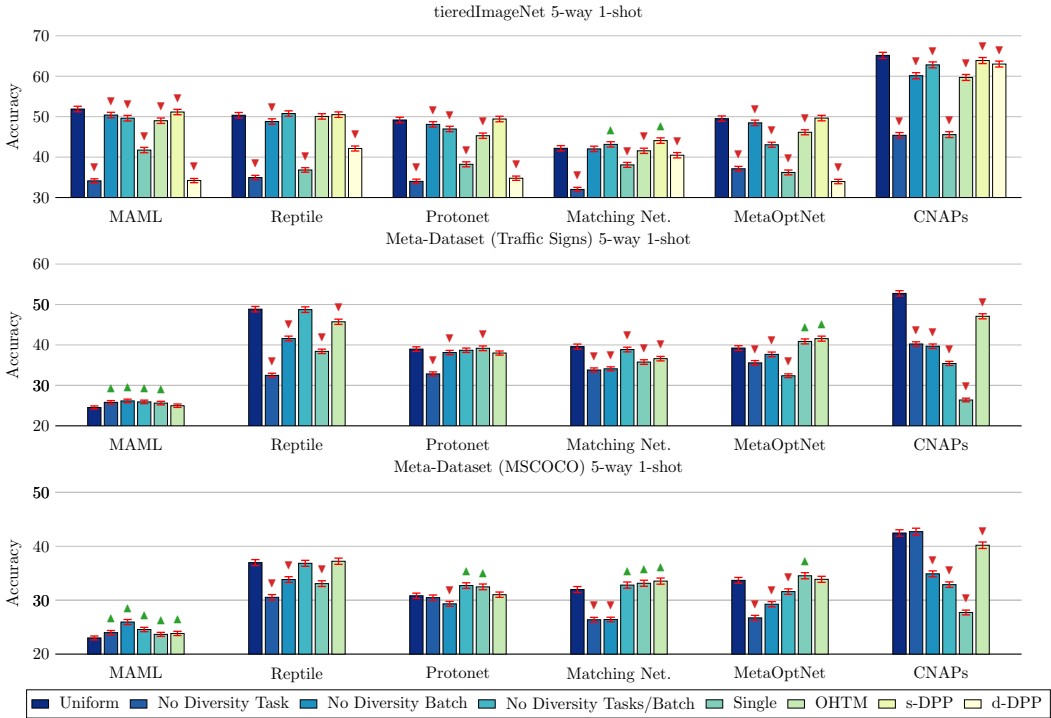

Figure 6: Average accuracy on *tiered*ImageNet 5-way 1-shot, Meta-Dataset Traffic Sign 5-way 1-shot & Meta-Dataset MSCOCO 5-way 1-shot, with a 95% confidence interval. We denote all samplers that are worse than the Uniform Sampler and are statistically significant (with a p-value $p = 0.05$) with ▼, and those that are significantly better than the Uniform Sampler with ▲.

Figure 6 presents the performance of the models on the Meta-Dataset Traffic Sign and Meta-Dataset MSCOCO datasets. We only present the results on Traffic Sign and MSCOCO of the Meta-Dataset, as these two sub-datasets are exclusively used for testing and are an accurate representation of the generalization power of the models when trained with different levels of task diversity. Other results on the Meta-Dataset are presented in Table 4. We empirically show that task diversity does not lead to any significant boost in the performance of the models. In the subsequent section, we discuss some of the other key findings from our work.

## 5 Discussion

In this section, we discuss few empirical results from our experiments and shed light on some of the key findings from our research.

**Poor performance by NDT Sampler** The lowest performance is consistently obtained by the No Diversity Task Sampler, which is reasonable since the model only sees one task throughout its training. What is fascinating is that just one task is sufficient for the model to reach a reasonably decent performance in most cases. We do notice instances where NDT Sampler performs very well on a few sub-datasets of the Meta-Dataset. This can be explained by the fact that the model has only be trained for a single sub-dataset and has relatively less class variability (noise) when compared to training on multiple sub-datasets.

**Poor performance by Single Batch Uniform Sampler** Consequently, the Single Batch Uniform Sampler does perform poorly on most datasets, including Omniglot, *mini*ImageNet, and *tiered*ImageNet. This is reasonable since the model is trained on a tiny pool of the dataset. However, we notice instances where the Sampler performs very well on a few sub-datasets of the Meta-Dataset. We hypothesize that training on fewer samples keeps the model unaware of the inherent

class variability generated by training on diverse datasets and aids better performance in the case of Meta-Dataset.

**Disparity between Single Batch Uniform and NDTB Sampler**   Another exciting result is the Disparity between Single Batch Uniform Sampler and No Diversity Tasks per Batch Sampler. As mentioned earlier, the only difference between the two samplers is that tasks are repeated in the latter. However, this repetition seems to offer a great deal of information to the model and allows the model to perform on par with the Uniform Sampler. It might be possible that the Single Batch Uniform Sampler obtains the performance observed by the No Diversity Tasks per Batch Sampler if trained for enough epochs. However, it would be safe to comment that the convergence of the model is significantly faster in the latter. Thus, repeating tasks might help speed up the convergence of the model when we have a fixed and handful amount of data. However, the same is not valid for models trained on Meta-Dataset. Although both samplers are trained over a similar pool of datasets, the NDTB sampler sees more data and might lead to more inherent class variability generated by training on diverse datasets. This might explain why repeating tasks leads to lower performance in this case.

**Disparity between s-DPP and d-DPP Sampler**   We also note that s-DPP and d-DPP samplers do not offer any boost in performance when compared to the regular Uniform Sampler. Furthermore, there seems to be a significant disparity between these two samplers. We believe that d-DPP, which computes the most diverse tasks at regular intervals, harms the model with the diverse tasks since we observe that the model's performance degrades over epochs. For example, consider the scenario where the model is trained on tasks involving dogs and tractors. This task is relatively easy to learn and would not require the model to fine-tune a great deal. However, during test time, suppose our task involves classifying cats and dogs; this would be a problem since the model has not learned the intricacies of the two classes. Thus, diversity seems to do more harm than good in this case. The best example of this is observed by Matching Networks in Omniglot 5-way 1-shot setting as shown in Figure 10, where each instance of diverse sampling harms the model significantly.

**Limitation of samplers with DPP backbone**   Samplers such as s-DPP and d-DPP, which use DPP to sample diverse tasks, require task embeddings of every class in the dataset. Computing these task embeddings, although intensive, might be sustainable for small datasets such as Omniglot, *mini*ImageNet, and *tiered*ImageNet. However, computing the task embedding for every class of a dataset as large as the Meta-Dataset is nearly impossible due to the time and memory constraints. Hence, in our experiments on the Meta-Dataset, we do not run the model using the s-DPP and d-DPP Sampler and only report the findings from the remaining samplers.

**OHTM Sampler offers no significant performance boost**   The OHTM Sampler is quite sophisticated since it regularly samples diverse tasks, as well as selects the most challenging tasks to improve the model. It is needless to say; the model requires more computational power and time than the Uniform Sampler. However, the OHTM Sampler offers no significant boost in performance when compared to the Uniform Sampler in the case of Omniglot, *mini*ImageNet, and *tiered*ImageNet. However, for Meta-Dataset, we notice that the OHTM Sampler sometimes leads to improved performance. This finding is quite puzzling since the Sampler works similarly across all datasets and does not address the inherent class variability generated from training on diverse datasets such as the Meta-Dataset. The behavior of the OHTM Sampler on Meta-Datasets warrants further research.

**Comparison between NDTB, NDB, and Uniform Sampler**   From our experiments, we also notice that the No Diversity Tasks per Batch Sampler and No Diversity Batch Sampler are pretty similar to the Uniform Sampler in terms of performance. This would suggest that the model trained on only a data fragment can perform similarly to that trained on the Uniform Sampler.

**Abnormal run of matching networks d-DPP (20-way 1-shot)**   In our run on the matching networks with the d-DPP Sampler under the 20-way 1-shot setting, we ran across a peculiar error. The prototypes generated by the matching networks were sometimes not fit to be used by the d-DPP Sampler to sample 20 unique classes. The reason is that the rank of the matrix generated using the embeddings was lower than the required number of classes per task (i.e., 20). To create a workaround for this sole experiment, we chose to sample 5 diverse classes at a time and append them

to create the task. We hypothesize that the prototypes created by matching networks are unsuitable for downstream tasks and warrant further research regarding this behavior.

**Poor performance of MAML on Meta-Dataset**   In our experiments on Meta-Dataset, we notice that MAML performs significantly worse than other models. Some of the reasons for this disparity of performance when compared to Triantafillou et al. (2019) have been discussed in detail in Appendix A.1. Furthermore, we hypothesize that the poor performance of MAML can be attributed to the way its adaptation process works: MAML learns within episode weights of the model before adapting to a set of new tasks via meta-update or outer loop update. This new task is again sampled from the pool of tasks and is a different set of data altogether. In most cases, where the model is trained on only one dataset, this intuition would make sense and lead to high-performing models. However, in the case of Meta-Dataset, where the new set of tasks might be of entirely different datasets or domains, this approach tends to do more harm to the model rather than aid. In the work of Triantafillou et al. (2019), the authors adapted MAML such that it focuses on learning the within-episode initialization $\theta$ of the embedding network so that it can be rapidly adapted for a new task. This allows the model to learn from a variable number of ways and shots per episode.

**Peculiar behavior with MetaOptNet model**   Compared to all other models, MetaOptNet seems to be immune to the effects of task diversity to a great extent. The convergence of the model seems to follow a general pattern and achieve similar performance across task distributions except for the Single Batch Uniform Sampler and No Diversity Task sampler. Furthermore, we do not observe the expected pattern of d-DPP Sampler, where the performance drops upon mining diverse tasks. We present the convergence graph of the MetaOptNet model on Omniglot 5-way 1-shot run in Figure 11 in the Appendix with an added smoothing factor of 1.

**General Trend**   From our experiments, we notice that there are generally two classes of samplers: High Performing Samplers and Low Performing Samplers. The High Performing Samplers include No Diversity Batch, No Diversity Tasks per Batch, Uniform, OHTM, and s-DPP Sampler. The Low Performing Samplers include No Diversity Task, Single Batch Uniform, and d-DPP Sampler. This trend is shared across all datasets and models. There are some perturbations in ranking within the two classes, but the High Performing Samplers tend to perform better than the Low Performing Samplers.

## 6   CONCLUSION

In this paper, we have studied the effect of task diversity in meta-learning. We have empirically shown that task diversity does not lead to any significant boost in performance in meta-learning. Instead, limiting task diversity and repeating the same tasks over the training phase allows us to obtain similar performances to the Uniform Sampler without any significant adverse effects. Furthermore, We also show that sophisticated samplers such as OHEM or DPP samplers do not offer any significant boost in performance. In contradiction, we notice that increasing task diversity using the d-DPP Sampler hampers the performance of the meta-learning model. Our experiments using the NDTB and NDB empirically show that a model trained on even a tiny data fragment can perform similarly to a model trained using Uniform Sampler. This is a crucial finding since this questions the need to increase the support set pool to improve the models' performance. We believe that the experiments we performed lay the roadwork to further research for the effect of task diversity domain in meta-learning and lay some groundwork and rules of thumb for task sampling for meta-learning.

REPRODUCABILITY STATEMENT

In this paper, we work with four different datasets - Omniglot, *mini*ImageNet, *tiered*ImageNet and Meta-Dataset. Additional details about setting up these datasets is available in Appendix A.1. Furthermore, we experiment with six different models - MAML, Reptile, Protonet, Matching Networks, MetaOptNet, and CNAPs. All these models were run after reproducing from their open-source codes. Additional details about setting up these models are available in Appendix A.2.

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

# A APPENDIX

## A.1 DATASETS

Omniglot, *mini*ImageNet and *tiered*ImageNet were extracted using pytorch-meta. The Meta-Dataset was downloaded using the setup information from the official repository: `https://github.com/google-research/meta-dataset`.

**Omniglot** Omniglot is a benchmark dataset proposed by Lake et al. (2011) for few-shot image classification tasks. Omniglot dataset consists of 20 instances and 1623 characters from 50 different alphabets. We experiment with both 5-way 1-shot and 2-way 1-shot in this work.

***mini*ImageNet** *mini*ImageNet is another benchmark dataset proposed by Ravi & Larochelle (2016) for few-shot image classification tasks. The *mini*ImageNet dataset involves 64 training classes, 12 validation classes, and 24 test classes. We run under the setting 5-way 1-shot for this experiment.

***tiered*ImageNet** *tiered*ImageNet is notorious as a difficult benchmark dataset proposed by Ren et al. (2018). The *tiered*ImageNet dataset is a larger subset of ILSVRC-12 with 608 classes (779,165 images) grouped into 34 higher-level nodes in the ImageNet human-curated hierarchy. This set of nodes is partitioned into 20, 6, and 8 disjoint sets of training, validation, and testing nodes, and the corresponding classes form the respective meta-sets. As argued in Ren et al. (2018), this split near the root of the ImageNet hierarchy results in a more challenging yet realistic regime with test classes that are less similar to training classes. We run under the setting 5-way 1-shot for this experiment.

**Meta-Dataset** Meta-Dataset is notorious as a difficult benchmark dataset proposed by Triantafillou et al. (2019). The Meta-Dataset is much larger than any previous benchmark and is comprised of multiple existing datasets. This invites research into how diverse data sources can be exploited by a meta-learner and allows us to evaluate a more challenging generalization problem to new datasets altogether. Specifically, Meta-Dataset leverages data from the following 10 datasets: ILSVRC-2012 (Russakovsky et al. (2015)), Omniglot (Lake et al. (2011)), Aircraft (Maji et al. (2013)), CUB-200-2011 (Wah et al. (2011)), Describable Textures (Cimpoi et al. (2014)), Quick Draw (Jongejan et al. (2016)), Fungi (Schroeder & Cui (2018)), VGG Flower (Nilsback & Zisserman (2008)), Traffic Signs (Houben et al. (2013)) and MSCOCO (Lin et al. (2014)). There exist few classes with fewer image samples than 16. This becomes an issue since we need 1 for training and 15 for testing. We repeat some of the support images for these classes to make up for the lack of examples. Since the number of such classes is minimal, we justify this use as this solution cannot significantly increase the model's performance. Furthermore, unlike previous experiments on this dataset which use a variable number of ways and shots during training, we train with a fixed number of ways and shots. Furthermore, unlike the works in Triantafillou et al. (2019), we do not perform any pre-training to help aid the model. We believe these are the main attributions for the disparity of our performance. However, since we are focused on the relative performance of the samplers for a given model, this discrepancy would not affect our study of task diversity in any manner. We again run under the setting 5-way 1-shot for this experiment.

## A.2 MODELS

This section describes some of the models we used for our experiments and the hyperparameters used for their training.

### A.2.1 PROTOTYPICAL NETWORKS

Prototypical Networks proposed by Snell et al. (2017) constructs a prototype for each class and then classifies each query example as the class whose prototype is 'nearest' to it under Euclidean distance. More concretely, the probability that a query example $x^*$ belongs to class $k$ is defined as:

$$p(y^* = k|x^*, \mathcal{S}) = \frac{\exp(-\|g(x^*) - c_k\|_2^2)}{\sum_{k' \in \{1,...,N\}} \exp(-\|g(x^*) - c_{k'}\|_2^2)} \tag{4}$$

Where $c_k$ is the 'prototype' for class $k$: the average embeddings of class $k$'s support examples.

HYPERPARAMETERS   In our experiments on Omniglot and *mini*ImageNet, and *tiered*ImageNet under a 5-way, 1-shot setting, we run the model for 100 epochs with a batch size of 32 and a meta-learning rate of 0.001. We use an Adam optimizer to make gradient steps and a StepLR scheduler with step size 0.4 and gamma 0.5. The same hyperparameters are used for training our model on Omniglot under a 20-way 1-shot setting.

We use the same parameters as the *mini*ImageNet to train our model on the Meta-Dataset under a 5-way 1-shot setting. However, we run with a batch size of 16 rather than 32 to accommodate the extensive training period and memory constraints.

### A.2.2   MATCHING NETWORKS

Matching Networks proposed by Vinyals et al. (2016) labels each query example as a cosine distance-weighted linear combination of the support labels:

$$p(y^* = k|x^*, \mathcal{S}) = \sum_{i=1}^{|\mathcal{S}|} \alpha(x^*, x_i)\Phi_{y_i=k}, \tag{5}$$

where $\Phi_A$ is the indicator function and $\alpha(x^*, x_i)$ is the cosine similarity between $g(x^*)$ and $g(x_i)$, softmax normalized over all support examples $x_i$, where $1 \leq i \leq |\mathcal{S}|$.

We had trouble reproducing the results from matching networks using cosine distance since the convergence seemed to be slow and the final performance dependent on the random initialization. This is similar to what is observed by other repositories such as https://github.com/oscarknagg/few-shot. Since we are focused on the relative performance of the samplers for a given model, this discrepancy would not affect our study of task diversity in any manner.

HYPERPARAMETERS   In our experiments on Omniglot, *mini*ImageNet, and *tiered*ImageNet under a 5-way, 1-shot setting, we run the model for 100 epochs with a batch size of 32 and an Adam optimizer with a meta-learning rate of 0.001 and a weight decay of 0.0001. The same hyperparameters are used for training our model on Omniglot under a 20-way 1-shot setting.

We use the same parameters as the *mini*ImageNet to train our model on the Meta-Dataset under a 5-way 1-shot setting. However, we run with a batch size of 16 rather than 32 to accommodate the extensive training period and memory constraints.

### A.2.3   MAML

MAML proposed by Finn et al. (2017) uses a linear layer parametrized by $\mathbf{W}$ and $\mathbf{b}$ on top of the embedding function $g(.; \theta)$ and classifies a query example as:

$$p(y^*|x^*, \mathcal{S}) = softmax(\mathbf{b}' + \mathbf{W}'g(x^*; \theta')), \tag{6}$$

where the output layer parameters $\mathbf{W}$' and $\mathbf{b}$' and the embedding function parameters $\theta'$ are obtained by performing a small number of within-episode training steps on the support set $\mathcal{S}$, starting from initial parameter values $(\mathbf{b}, \mathbf{W}, \theta)$.

HYPERPARAMETERS   In our experiments on Omniglot, *mini*ImageNet, and *tiered*ImageNet under a 5-way, 1-shot setting, we run the model for 150 epochs with a batch size of 32, with the Adam optimizer with a meta-learning rate of 0.001, number of inner adaptations as 1, and step size 0.4. For our experiments on Omniglot under the 20-way 1-shot setting, we set the step size of 0.1 and the number of inner adaptations to 5, batch size of 16, and kept all other hyperparameters constant.

We use the same parameters as the *mini*ImageNet to train our model on the Meta-Dataset under a 5-way 1-shot setting. However, we ran with a batch size of 16 rather than 32 and only trained for 100 epochs to accommodate the extensive training period and memory constraints.

### A.2.4   REPTILE

Like MAML, Reptile proposed by Nichol et al. (2018) learns an initialization for the parameters of a neural network model, such that when we optimize these parameters at test time, learning is fast -

i.e., the model generalizes from a small number of test tasks. Reptile converges towards a solution $\phi$ that is close (in Euclidean distance) to each task $\tau$'s manifold of optimal solutions. Let $\phi$ denote the network initialization, and $W = \phi + \Delta\phi$ denote the network weights after performing some sort of update. Let $\mathcal{W}_\tau^*$ denote the set of optimal network weights for task $\tau$. We want to find $\phi$ such that the distance $D(\phi, \mathcal{W}_\tau^*)$ is small for all tasks:

$$\min_\phi \mathbb{E}_\tau [\frac{1}{2} D(\phi, \mathcal{W}_\tau^*)^2] \tag{7}$$

The official repository seems to train the model with a 5-way 15-shot and test the model on a 5-way 1-shot. However, we do not consider this to be an accurate study for the effect of task diversity. In our work, we train and test the model in a 5-way 1-shot setting to ensure fair and accurate comparison with other models. We believe this to be the source of discrepancy in our performance scores. Since we are focused on the relative performance of the samplers for a given model, this discrepancy would not affect our study of task diversity in any manner.

HYPERPARAMETERS    In our experiments on Omniglot and *mini*ImageNet under a 5-way, 1-shot setting, we run the model for 150 epochs with a batch size of 32, a learning rate of 0.01, a meta-learning rate of 0.001, and a number of inner adaptations as 5. We use the SGD optimizer of inner steps and Adam optimizer for the outer steps. For our experiments on *tiered*ImageNet, we change the number of inner adaptations to 10, keeping all other hyperparameters constant. For our experiments on Omniglot under the 20-way 1-shot setting, we set the meta-learning rate to 0.0005 and the number of inner adaptations to 10 and kept all other hyperparameters constant. Furthermore, we only run the model for 50 epochs due to the very high training time.

We use the same parameters as the *tiered*ImageNet to train our model on the Meta-Dataset under a 5-way 1-shot setting. However, we run with a batch size of 16 rather than 32 to accommodate the extensive training period and memory constraints.

### A.2.5   CNAPS

Conditional Neural Adaptive Processes proposed by Requeima et al. (2019) is able to efficiently solve new multi-class classification problems after an initial training phase. The proposed approach, based on Conditional Neural Processes (CNPs) mentioned in Garnelo et al. (2018), adapts a small number of task-specific parameters for each new task encountered at test time. These parameters are conditioned on a set of training examples for the new task. They do not require any additional tuning to adapt both the final classification layer and feature extraction process. This allows the model to handle various input distributions. The CNPs construct predictive distributions given $x^*$ as:

$$p(y^*|x^*, \theta, D^\tau) = p(y^*|x^*, \theta, \Psi^\tau = \Psi_\phi(D^\tau)), \tag{8}$$

where $\theta$ are global classifer parameters shared across tasks, $\Psi^\tau$ are local task-specific parameters, produced by a function $\Psi_\phi(.)$ that acts of $D^\tau$. $\Psi_\phi(.)$ has another set of global parameters $\phi$ called *adaptation network parameters*. $\theta$ and $\phi$ are the learnable parameters in the model.

HYPERPARAMETERS    In all our experiments with CNAPs, we run the model for ten epochs with a batch size of 16 and a meta-learning rate of 0.01.

### A.2.6   METAOPTNET

MetaOptNet proposed by Lee et al. (2019) proposes a linear classifier as the base learner for a meta-learning based approach for few-shot learning. The approach uses a linear support vector machine (SVM) to learn a classifier given a set of labeled training examples. The generalization error is computed on a novel set of examples from the same task. The objective is to learn an embedding model $\phi$ that minimizes generalization (or test) error across tasks given a base learner $\mathcal{A}$. Formally, the learning objective is:

$$\min_\phi \mathbb{E}_\mathcal{T}[\mathcal{L}^{meta}(\mathcal{D}^{test}; \theta, \phi), \text{ where } \theta = \mathcal{A}(\mathcal{D}^{test}; \phi)]. \tag{9}$$

The choice of base learner $\mathcal{A}$ has a significant impact on the above equation. The base learner that computes $\theta = \mathcal{A}(\mathcal{D}^{test}; \phi)$ has to be efficient since the expectation has to be computed over a

distribution of tasks. This work considers base learners based on multi-class linear classifiers such as SVM, where the base learner's object is convex. Thus, the base learner can be simplified as:

$$\theta = \mathcal{A}(\mathcal{D}^{test}; \phi) = \underset{\{\mathbf{w}_k\}}{\arg\min} \min_{\xi_i} \frac{1}{2} \sum_k \|\mathbf{w}_k\|_2^2 + C \sum_n \xi_n; \text{ subject to:}$$
$$\mathbf{w}_{y_n}.f_\phi(x_n) - \mathbf{w}_k.f_\phi(x_n) \geq 1 - \delta_{y_n,k} - \xi_n, \forall n, k \tag{10}$$

where $\mathcal{D}^{train} = \{(x_n, y_n)\}$, $C$ is the regularization parameter and $\delta_{.,.}$ is the Kronecker delta function.

Furthermore, the official repository seems to train the model with a 5-way 15-shot and test the model on a 5-way 1-shot. However, we do not consider this to be an accurate study for the effect of task diversity. In our work, we train and test the model in a 5-way 1-shot setting to ensure fair and accurate comparison with other models. We believe this to be the source of discrepancy in our performance scores. Since we are focused on the relative performance of the samplers for a given model, this discrepancy would not affect our study of task diversity in any manner.

HYPERPARAMETERS   In our experiments on Omniglot, *mini*ImageNet and *tiered*ImageNet under a 5-way, 1-shot setting, we run the model for 60 epochs with a batch size of 32 and a meta-learning rate of 0.01. We use an SGD optimizer with a momentum of 0.9 and a weight decay of 0.0001 to make gradient steps. We also use a LambdaLR scheduler to train our model. The same hyperparameters are used for training our model on Omniglot under a 20-way 1-shot setting.

We use the same parameters as the *mini*ImageNet to train our model on the Meta-Dataset under a 5-way 1-shot setting. However, we run with a batch size of 16 rather than 32 to accommodate the extensive training period and memory constraints.

## A.3   ADDITIONAL RESULTS

### A.3.1   RESULTS ON REGRESSION

We further experiment on the few-shot regression regime to maintain our findings on classification. For this problem, we trained on three few-shot regression datasets: (i) Sinusoid (Finn et al. (2017)), (ii)Sinusoid & Line (Finn et al. (2018)), and (iii) Harmonic (Lacoste et al. (2018). Table 1 presents our results using the mean square error metric on the 5-shot and 10-shot setting for all three datasets. Figure 7 and Figure 8 presents our results on the Sinusoid dataset specifically for easier comparison. For both MAML and Reptile, we use the same network, with two hidden layers, each of size 40.

HYPERPARAMETERS   For both MAML and Reptile, we train the model for 150 epochs, and use the same hyperparameters as Omniglot. Only in the case of MAML, we use a step size of 0.001 for Sinusoid and Sinusoid & Line dataset, instead of the traditional 0.4.

**Results**   We again observe that samplers that limit diversity, such as NDB, NDT/B samplers achieve performances similar to the uniform sampler with no adverse effects. Astonishingly, we also notice that the NDT sampler seems to outperform all other samplers in the regression domain. Furthermore, similar to our previous findings, samplers that increase diversity, such as the OHTM sampler, do not perform very well in this domain either, as shown in Figure 7 and Figure 8. We cannot train using DPP-oriented samplers for regression since the data is continuous, and computing features or representations for the entire range would be theoretically impossible.

### A.3.2   RESULTS ON CLASSIFICATION

In this section, we present the results with higher precision from our earlier experiments in a Table 2 and Table 3. Subsequently, we also plot convergence curves to aid better visualizations of findings mentioned earlier in Figure 10 and Figure 11. We also present the results of our models on the Meta-Dataset in Table 4.

**Statistical Results**   We compare the performance of different models to the Uniform Sampler. All samplers are poorer than the Uniform Sampler and are statistically significant with a confidence interval of 95%. We use the symbol † to represent the instances where the results are not statistically

| Dataset | Sampler | 5-shot | | 10-shot | |
|---|---|---|---|---|---|
| | | *MAML* | *Reptile* | *MAML* | *Reptile* |
| **Sinusoid** | Uniform Sampler | $0.94 \pm 0.06$ | $0.37 \pm 0.04$ | **0.50 ± 0.03** | $0.12 \pm 0.01$ |
| | No Diversity Task Sampler | $0.79 \pm 0.06$ ‡ | **0.35 ± 0.35** | $0.54 \pm 0.04$ † | **0.12 ± 0.01** † |
| | No Diversity Batch Sampler | **0.77 ± 0.05** ‡ | $0.37 \pm 0.04$ † | $0.56 \pm 0.04$ ‡ | $0.13 \pm 0.01$ † |
| | No Diversity Tasks per Batch Sampler | $0.92 \pm 0.06$ † | $0.37 \pm 0.04$ † | $0.52 \pm 0.04$ † | $0.12 \pm 0.01$ † |
| | Single Batch Uniform Sampler | $1.94 \pm 0.11$ | $0.71 \pm 0.06$ | $1.39 \pm 0.07$ | $0.25 \pm 0.02$ |
| | OHTM Sampler | $1.17 \pm 0.08$ | $0.84 \pm 0.30$ | $0.79 \pm 0.05$ | $0.38 \pm 0.03$ |
| **Sinusoid and Line** | Uniform Sampler | $4.08 \pm 0.32$ | $3.74 \pm 3.07$ | $2.78 \pm 0.22$ | $0.66 \pm 0.13$ |
| | No Diversity Task Sampler | **3.91 ± 0.32** † | **2.80 ± 0.30** † | $2.70 \pm 0.23$ † | $0.70 \pm 0.13$ † |
| | No Diversity Batch Sampler | $4.07 \pm 0.34$ † | $2.18 \pm 0.27$ † | **2.66 ± 0.22** † | $0.69 \pm 0.11$ † |
| | No Diversity Tasks per Batch Sampler | $4.17 \pm 0.37$ † | $2.82 \pm 0.31$ † | $2.73 \pm 0.22$ † | **0.55 ± 0.08** † |
| | Single Batch Uniform Sampler | $6.62 \pm 0.57$ | $5.52 \pm 4.45$ | $8.48 \pm 0.94$ | $1.19 \pm 0.15$ |
| | OHTM Sampler | $4.43 \pm 0.33$ | $3.67 \pm 0.51$ | $3.36 \pm 0.28$ | $1.19 \pm 0.15$ |
| **Harmonic** | Uniform Sampler | **1.07 ± 0.07** | $1.20 \pm 0.08$ | $1.07 \pm 0.07$ | $1.05 \pm 0.07$ |
| | No Diversity Task Sampler | $1.12 \pm 0.07$ † | $1.22 \pm 0.09$ † | $1.07 \pm 0.07$ † | $1.07 \pm 0.07$ † |
| | No Diversity Batch Sampler | $1.11 \pm 0.07$ † | $1.18 \pm 0.09$ † | **1.03 ± 0.06** † | $1.08 \pm 0.07$ † |
| | No Diversity Tasks per Batch Sampler | $1.14 \pm 0.07$ † | $1.18 \pm 0.08$ † | $1.05 \pm 0.07$ † | **1.05 ± 0.07** † |
| | Single Batch Uniform Sampler | $1.19 \pm 0.08$ | **1.10 ± 0.08** † | $1.06 \pm 0.07$ | $1.08 \pm 0.07$ † |
| | OHTM Sampler | $1.13 \pm 0.08$ | $1.18 \pm 0.08$ † | $1.42 \pm 0.15$ | $1.10 \pm 0.07$ † |

Table 1: Performance metric of our models on different task samplers in the few-shot regression setting.

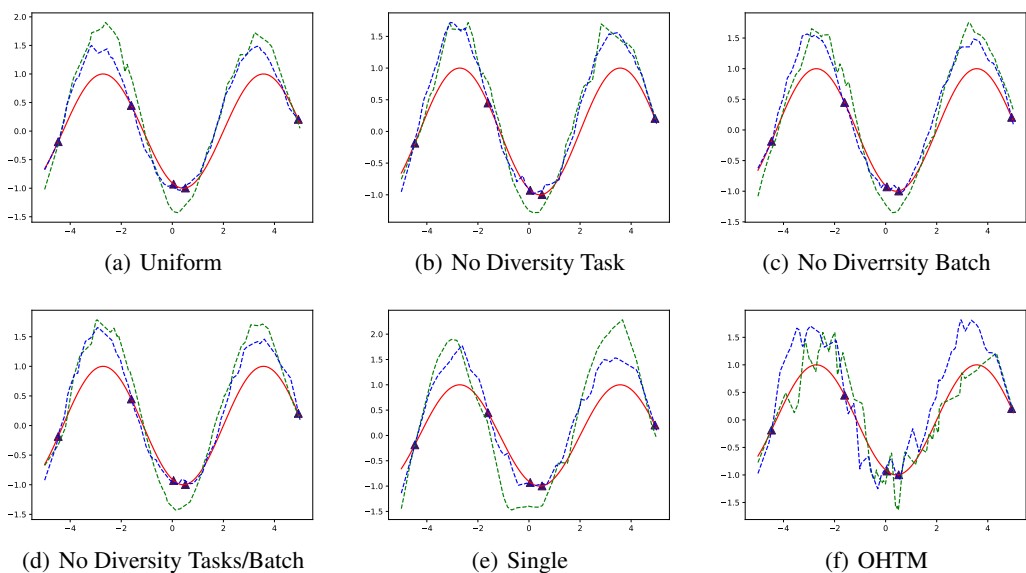

(a) Uniform  (b) No Diversity Task  (c) No Diverrsity Batch

(d) No Diversity Tasks/Batch  (e) Single  (f) OHTM

Figure 7: Few-shot adaptation for the simple regression task on Sinusoid 5-shot dataset. The ground truth is denoted by "——". The predicted output of MAML and Reptile are denoted by "‒ ‒" and "‒ ‒" respectively. The training points used for computing gradients is denoted by ▲.

significant and similar to the performance achieved by uniform Sampler. We only observe a few cases where a sampler performs significantly better than the Uniform Sampler, which we represent using the symbol ‡. To assess statistical significance, we used a paired-difference t-test, with a p-value $p = 0.05$.

**Comparisons with SBU Sampler**    Previously, we compared the performance of NDTB and SBU samplers. However, due to the possibility of unfair comparisons, limited by the number of iterations of the SBU sampler, we extend the SBU Sampler to propose two new samplers: (1) **SBU-unbounded sampler** - This sampler is similar to the SBU sampler but is run for more iterations. However, this sampler allows the model to be trained on more classes than our traditional SBU sampler. Since this remains an unfair comparison, we use this as a higher bound/ideal performance and propose a more appropriate sampler. (2) **SBU-bounded sampler** - This sampler is again similar to the SBU sampler but ran for more iterations with a bounded pool of tasks, similar to the NDTB

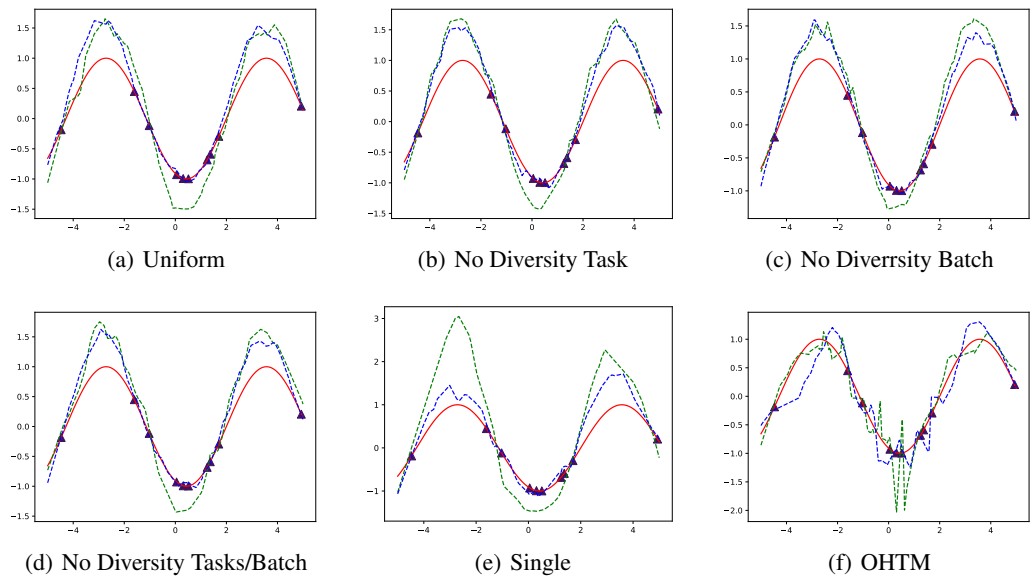

(a) Uniform      (b) No Diversity Task      (c) No Diverrsity Batch

(d) No Diversity Tasks/Batch      (e) Single      (f) OHTM

Figure 8: Few-shot adaptation for the simple regression task on Sinusoid 10-shot dataset. The ground truth is denoted by "——". The predicted output of MAML and Reptile are denoted by "---" and "---" respectively. The training points used for computing gradients is denoted by ▲.

sampler or the SBU sampler. Our results from these experiments are presented in Figure 9. Table 5 shows the same results with higher precision.

We notice that training for more iterations using samplers such as SBU-unbounded and SBU-bounded leads to a boost in performance. However, we also observe that the NDTB sampler remains to perform better in most cases. With this, we maintain our previous findings. Furthermore, we also notice a peculiar behavior where the SBU-bounded sampler performs better than the SBU-unbounded sampler in most cases. This finding is interesting because the SBU-unbounded sampler has access to more data, and the model being trained should lead to better representations and performance. However, we notice that repeating the tasks and fixing the number of classes achieves the same performance with little to no adverse effects.

| Dataset | Sampler | MAML | Reptile | Protonet | Matching Networks | MetaOptNet | CNAPs |
|---|---|---|---|---|---|---|---|
| Omniglot | Uniform Sampler | **98.38 ± 0.17** | 94.64 ± 0.32 | **97.82 ± 0.23** | 94.71 ± 0.39 | **98.04 ± 0.22** | **95.01 ± 0.35** |
| | No Diversity Task Sampler | 85.46 ± 0.59 | 81.59 ± 0.57 | 84.55 ± 0.56 | 64.41 ± 0.74 | 84.15 ± 0.57 | 62.06 ± 0.83 |
| | No Diversity Batch Sampler | 97.17 ± 0.25 | 93.83 ± 0.34 | 96.67 ± 0.27 | 76.10 ± 0.65 | 97.11 ± 0.26 | 91.07 ± 0.46 |
| | No Diversity Tasks per Batch Sampler | 97.76 ± 0.20 | 94.55 ± 0.31 † | 97.18 ± 0.25 | 93.97 ± 0.40 | 96.80 ± 0.27 | 90.84 ± 0.47 |
| | Single Batch Uniform Sampler | 93.84 ± 0.37 | 92.60 ± 0.38 | 95.95 ± 0.31 | 92.98 ± 0.44 | 95.76 ± 0.31 | 75.86 ± 0.73 |
| | OHTM Sampler | 97.74 ± 0.20 | 93.89 ± 0.34 | 97.22 ± 0.25 | 93.48 ± 0.43 | 96.12 ± 0.29 | 91.51 ± 0.47 |
| | s-DPP Sampler | 97.61 ± 0.21 | **94.79 ± 0.30** † | 97.22 ± 0.24 | 92.29 ± 0.44 | 95.83 ± 0.30 | 95.00 ± 0.33 † |
| | d-DPP Sampler | 92.43 ± 0.42 | 92.60 ± 0.38 | 95.59 ± 0.33 | 93.67 ± 0.41 | 94.08 ± 0.35 | 94.01 ± 0.41 |
| MiniImagenet | Uniform Sampler | **48.86 ± 0.62** | 41.42 ± 0.56 | **48.56 ± 0.60** | 43.84 ± 0.58 | **55.02 ± 0.66** | **64.48 ± 0.71** |
| | No Diversity Task Sampler | 36.70 ± 0.53 | 32.38 ± 0.48 | 37.83 ± 0.53 | 35.08 ± 0.53 | 36.62 ± 0.55 | 46.51 ± 0.63 |
| | No Diversity Batch Sampler | 48.78 ± 0.60 † | 40.80 ± 0.54 | 47.32 ± 0.62 | 42.15 ± 0.58 | 53.50 ± 0.63 | 60.92 ± 0.68 |
| | No Diversity Tasks per Batch Sampler | 48.17 ± 0.62 | **41.49 ± 0.56** † | 47.73 ± 0.60 | 42.54 ± 0.53 | 50.60 ± 0.62 | 64.11 ± 0.68 † |
| | Single Batch Uniform Sampler | 41.76 ± 0.56 | 22.96 ± 0.33 | 41.35 ± 0.56 | 40.00 ± 0.54 | 39.10 ± 0.54 | 45.47 ± 0.67 |
| | OHTM Sampler | 48.30 ± 0.58 | 40.44 ± 0.54 | 47.45 ± 0.59 | 43.05 ± 0.55 | 47.11 ± 0.58 | 59.62 ± 0.69 |
| | s-DPP Sampler | 48.14 ± 0.59 | 40.19 ± 0.56 | 47.22 ± 0.58 | 42.66 ± 0.56 | 52.74 ± 0.63 | 63.26 ± 0.69 |
| | d-DPP Sampler | 35.61 ± 0.50 | 35.91 ± 0.54 | 34.77 ± 0.50 | 43.15 ± 0.58 | 34.67 ± 0.53 | 60.96 ± 0.70 |
| Tiered-Imagenet | Uniform Sampler | **51.89 ± 0.68** | 50.35 ± 0.69 | 49.18 ± 0.68 | 42.18 ± 0.66 | 49.51 ± 0.67 | **65.16 ± 0.75** |
| | No Diversity Task Sampler | 34.13 ± 0.51 | 34.94 ± 0.54 | 34.01 ± 0.54 | 32.01 ± 0.51 | 37.12 ± 0.59 | 45.40 ± 0.68 |
| | No Diversity Batch Sampler | 50.40 ± 0.68 | 48.80 ± 0.67 | 48.09 ± 0.67 | 42.03 ± 0.67 † | 48.47 ± 0.67 | 60.13 ± 0.75 |
| | No Diversity Tasks per Batch Sampler | 49.62 ± 0.69 | **50.77 ± 0.67** † | 46.96 ± 0.67 | 43.15 ± 0.66 ‡ | 43.05 ± 0.63 | 62.81 ± 0.73 |
| | Single Batch Uniform Sampler | 41.74 ± 0.66 | 36.82 ± 0.56 | 38.22 ± 0.62 | 38.08 ± 0.61 | 36.21 ± 0.60 | 45.56 ± 0.72 |
| | OHTM Sampler | 49.00 ± 0.67 | 50.06 ± 0.67 † | 45.29 ± 0.66 | 41.56 ± 0.65 | 46.14 ± 0.62 | 59.70 ± 0.74 |
| | s-DPP Sampler | 51.15 ± 0.66 | 50.50 ± 0.68 † | **49.42 ± 0.68** † | **44.08 ± 0.68** ‡ | 49.65 ± 0.68 † | 63.88 ± 0.75 |
| | d-DPP Sampler | 34.20 ± 0.52 | 42.13 ± 0.62 | 34.78 ± 0.53 | 40.47 ± 0.66 | 33.96 ± 0.54 | 63.01 ± 0.73 |

Table 2: Performance metric of our models on different task samplers in the 5-way 1-shot setting.

| Dataset | Sampler | MAML | Reptile | Protonet | Matching Networks | MetaOptNet | CNAPs |
|---|---|---|---|---|---|---|---|
| Omniglot | Uniform Sampler | **91.28 ± 0.22** | 90.09 ± 0.22 | 93.72 ± 0.20 | **74.62 ± 0.38** | 90.20 ± 0.23 | **92.09 ± 0.22** |
| | No Diversity Task Sampler | 83.39 ± 0.29 | 59.49 ± 0.33 | 85.84 ± 0.27 | 26.50 ± 0.32 | 88.40 ± 0.26 | 73.82 ± 0.39 |
| | No Diversity Batch Sampler | 89.07 ± 0.25 | 88.23 ± 0.23 | 93.18 ± 0.20 | 71.77 ± 0.38 | 91.24 ± 0.22 | 89.56 ± 0.24 |
| | No Diversity Tasks per Batch Sampler | 90.77 ± 0.23 | **91.15 ± 0.21** | **93.85 ± 0.19** † | 61.31 ± 0.41 | 89.59 ± 0.24 | 89.99 ± 0.24 |
| | Single Batch Uniform Sampler | 82.45 ± 0.31 | 80.89 ± 0.27 | 92.67 ± 0.20 | 54.01 ± 0.40 | 70.81 ± 0.35 | 77.54 ± 0.37 |
| | OHTM Sampler | 91.25 ± 0.22 † | 89.92 ± 0.22 | 93.33 ± 0.20 | 72.20 ± 0.38 | **91.56 ± 0.23** | 89.51 ± 0.25 |
| | s-DPP Sampler | 88.79 ± 0.24 | 85.40 ± 0.25 | 90.90 ± 0.22 | 72.86 ± 0.37 | 91.47 ± 0.22 | 90.98 ± 0.22 |
| | d-DPP Sampler | 87.45 ± 0.26 | 83.38 ± 0.27 | 91.21 ± 0.22 | 70.84 ± 0.38 | 90.40 ± 0.24 † | 90.11 ± 0.24 |

Table 3: Performance metric of our models on different task samplers in the 20-way 1-shot setting.

| Dataset | Sampler | MAML | Reptile | Protonet | Matching Networks | MetaOptNet | CNAPs |
|---|---|---|---|---|---|---|---|
| Meta-Dataset (ILSVRC) | Uniform Sampler | 22.24 ± 0.33 | 31.32 ± 0.47 | 25.41 ± 0.35 | 27.52 ± 0.45 | 28.74 ± 0.43 | 40.11 ± 0.59 |
| | No Diversity Task Sampler | 23.02 ± 0.35 | 25.43 ± 0.40 | 26.68 ± 0.44 | 24.32 ± 0.37 | 23.85 ± 0.36 | **42.23 ± 0.64** |
| | No Diversity Batch Sampler | **23.24 ± 0.36** | 28.70 ± 0.44 | 24.71 ± 0.34 | 24.25 ± 0.37 | 25.64 ± 0.40 | 34.07 ± 0.52 |
| | No Diversity Tasks per Batch Sampler | 23.14 ± 0.37 | 31.25 ± 0.51 † | 27.04 ± 0.42 | 26.47 ± 0.44 | 27.01 ± 0.43 | 33.00 ± 0.53 |
| | Single Batch Uniform Sampler | 22.99 ± 0.34 | 26.43 ± 0.41 | **27.05 ± 0.42** | 27.79 ± 0.44 † | 28.00 ± 0.45 | 26.18 ± 0.44 |
| | OHTM Sampler | 22.66 ± 0.35 | **31.62 ± 0.48** | 26.48 ± 0.39 | 27.05 ± 0.43 † | **29.03 ± 0.44** † | 38.12 ± 0.57 |
| Meta-Dataset (Omniglot) | Uniform Sampler | 30.93 ± 0.69 | 86.07 ± 0.50 | **77.77 ± 0.65** | **77.61 ± 0.64** | **81.82 ± 0.60** | **87.08 ± 0.52** |
| | No Diversity Task Sampler | 31.17 ± 0.65 † | 41.36 ± 0.74 | 61.26 ± 0.73 | 60.59 ± 0.69 | 73.67 ± 0.65 | 60.18 ± 0.73 |
| | No Diversity Batch Sampler | **32.70 ± 0.75** | 72.13 ± 0.66 | 70.86 ± 0.67 | 56.61 ± 0.71 | 73.27 ± 0.64 | 73.69 ± 0.69 |
| | No Diversity Tasks per Batch Sampler | 30.36 ± 0.71 † | **86.29 ± 0.49** † | 72.33 ± 0.69 | 72.96 ± 0.68 | 63.17 ± 0.70 | 56.14 ± 0.72 |
| | Single Batch Uniform Sampler | 30.21 ± 0.74 † | 75.39 ± 0.66 | 63.52 ± 0.69 | 72.95 ± 0.71 | 73.44 ± 0.66 | 42.72 ± 0.74 |
| | OHTM Sampler | 31.91 ± 0.70 ‡ | 83.42 ± 0.53 | 76.55 ± 0.65 | 74.93 ± 0.66 | 74.56 ± 0.66 | 78.00 ± 0.69 |
| Meta-Dataset (Aircraft) | Uniform Sampler | 22.52 ± 0.34 | 40.62 ± 0.54 | 28.34 ± 0.41 | 27.93 ± 0.41 | **32.12 ± 0.47** | **38.64 ± 0.54** |
| | No Diversity Task Sampler | 22.76 ± 0.32 † | 24.55 ± 0.35 | 26.77 ± 0.36 | 22.58 ± 0.30 | 24.73 ± 0.36 | 30.37 ± 0.40 |
| | No Diversity Batch Sampler | 22.32 ± 0.31 ‡ | 33.11 ± 0.48 | **28.57 ± 0.40** † | 23.63 ± 0.33 | 25.76 ± 0.36 | 28.84 ± 0.39 |
| | No Diversity Tasks per Batch Sampler | 23.00 ± 0.33 ‡ | 40.59 ± 0.52 † | 28.32 ± 0.41 † | 27.98 ± 0.42 † | 27.13 ± 0.40 | 26.09 ± 0.36 |
| | Single Batch Uniform Sampler | 22.40 ± 0.31 ‡ | 29.19 ± 0.40 | 28.51 ± 0.40 † | 26.47 ± 0.39 | 27.53 ± 0.42 | 21.86 ± 0.26 |
| | OHTM Sampler | **23.34 ± 0.32** | 38.73 ± 0.54 | 27.99 ± 0.38 † | **29.53 ± 0.45** ‡ | 31.08 ± 0.47 | 35.96 ± 0.50 |
| Meta-Dataset (Birds) | Uniform Sampler | 22.30 ± 0.32 | **47.79 ± 0.61** | 29.84 ± 0.42 | 34.36 ± 0.53 | 33.70 ± 0.48 | **47.43 ± 0.62** |
| | No Diversity Task Sampler | 23.13 ± 0.33 † | 27.25 ± 0.38 | 27.08 ± 0.38 | 25.95 ± 0.39 | 26.78 ± 0.41 | 40.42 ± 0.61 |
| | No Diversity Batch Sampler | **25.31 ± 0.41** | 36.66 ± 0.51 | 28.47 ± 0.38 | 26.91 ± 0.41 | 28.55 ± 0.40 | 31.95 ± 0.48 |
| | No Diversity Tasks per Batch Sampler | 23.93 ± 0.37 ‡ | 45.36 ± 0.56 | 29.48 ± 0.43 † | 32.11 ± 0.48 | 27.86 ± 0.41 | 31.28 ± 0.47 |
| | Single Batch Uniform Sampler | 23.69 ± 0.36 ‡ | 33.70 ± 0.47 | 30.40 ± 0.46 † | 32.39 ± 0.50 | 30.80 ± 0.44 | 25.13 ± 0.38 |
| | OHTM Sampler | 23.22 ± 0.35 | 46.38 ± 0.61 | **30.83 ± 0.43** ‡ | **35.17 ± 0.54** ‡ | **37.18 ± 0.50** ‡ | 43.77 ± 0.59 |
| Meta-Dataset (Textures) | Uniform Sampler | 22.51 ± 0.34 | **33.33 ± 0.48** | 26.63 ± 0.37 | 28.74 ± 0.42 | 27.44 ± 0.40 | **38.10 ± 0.50** |
| | No Diversity Task Sampler | 22.46 ± 0.35 † | 24.93 ± 0.37 | 24.95 ± 0.37 | 23.58 ± 0.34 | 23.80 ± 0.34 | 33.11 ± 0.51 |
| | No Diversity Batch Sampler | 22.32 ± 0.32 † | 29.40 ± 0.44 | 26.80 ± 0.37 † | 23.84 ± 0.34 | 24.95 ± 0.35 | 32.46 ± 0.49 |
| | No Diversity Tasks per Batch Sampler | 22.69 ± 0.35 † | 31.94 ± 0.45 | 26.53 ± 0.37 † | 25.65 ± 0.37 | 23.99 ± 0.34 | 31.77 ± 0.44 |
| | Single Batch Uniform Sampler | 21.51 ± 0.29 | 26.97 ± 0.39 | 27.08 ± 0.37 † | 25.52 ± 0.38 | 26.29 ± 0.38 | 22.93 ± 0.35 |
| | OHTM Sampler | **22.94 ± 0.36** † | 31.67 ± 0.48 | **28.33 ± 0.40** † | **29.36 ± 0.42** ‡ | **28.34 ± 0.41** ‡ | 34.90 ± 0.46 |
| Meta-Dataset (Quick Draw) | Uniform Sampler | 34.84 ± 0.63 | 55.31 ± 0.69 | **51.80 ± 0.67** | **52.61 ± 0.68** | **56.17 ± 0.67** | **58.54 ± 0.70** |
| | No Diversity Task Sampler | 34.95 ± 0.62 † | 37.59 ± 0.58 | 48.94 ± 0.64 | 42.82 ± 0.62 | 51.42 ± 0.68 | 44.04 ± 0.64 |
| | No Diversity Batch Sampler | 35.45 ± 0.63 † | 46.98 ± 0.64 | 44.28 ± 0.63 | 36.60 ± 0.55 | 48.00 ± 0.63 | 47.53 ± 0.68 |
| | No Diversity Tasks per Batch Sampler | 35.31 ± 0.61 † | **55.49 ± 0.67** | 50.77 ± 0.65 | 47.41 ± 0.67 | 48.96 ± 0.63 | 41.24 ± 0.63 |
| | Single Batch Uniform Sampler | **36.58 ± 0.62** ‡ | 50.10 ± 0.64 | 48.52 ± 0.66 | 48.02 ± 0.68 | 51.96 ± 0.67 | 29.29 ± 0.48 |
| | OHTM Sampler | 35.13 ± 0.61 † | 53.42 ± 0.68 | 49.90 ± 0.69 | 50.59 ± 0.68 | 54.11 ± 0.65 | 51.99 ± 0.68 |
| Meta-Dataset (Fungi) | Uniform Sampler | 23.04 ± 0.36 | **39.78 ± 0.57** | 29.43 ± 0.42 | 34.45 ± 0.54 | 34.65 ± 0.53 | 40.37 ± 0.58 |
| | No Diversity Task Sampler | 23.53 ± 0.36 † | 29.22 ± 0.45 | 27.69 ± 0.40 | 24.57 ± 0.35 | 27.25 ± 0.43 | 35.26 ± 0.50 |
| | No Diversity Batch Sampler | **25.05 ± 0.41** | 34.84 ± 0.54 | 28.43 ± 0.42 | 25.97 ± 0.41 | 30.46 ± 0.50 | 30.75 ± 0.46 |
| | No Diversity Tasks per Batch Sampler | 24.47 ± 0.41 ‡ | 38.80 ± 0.59 | 30.38 ± 0.46 ‡ | 32.03 ± 0.54 | 28.82 ± 0.44 | 31.26 ± 0.47 |
| | Single Batch Uniform Sampler | 23.28 ± 0.30 ‡ | 31.28 ± 0.49 | 30.12 ± 0.46 ‡ | 31.94 ± 0.51 | 30.49 ± 0.48 | 25.90 ± 0.43 |
| | OHTM Sampler | 23.88 ± 0.37 ‡ | 34.84 ± 0.54 | **31.13 ± 0.47** ‡ | **35.63 ± 0.56** ‡ | **36.40 ± 0.56** ‡ | **41.85 ± 0.57** ‡ |
| Meta-Dataset (VGG Flower) | Uniform Sampler | 30.28 ± 0.51 | 65.76 ± 0.64 | 50.98 ± 0.55 | **58.64 ± 0.71** | 58.28 ± 0.67 | **65.12 ± 0.68** |
| | No Diversity Task Sampler | 31.19 ± 0.49 † | 49.91 ± 0.59 | 45.36 ± 0.57 | 45.50 ± 0.64 | 37.46 ± 0.56 | 52.21 ± 0.65 |
| | No Diversity Batch Sampler | **35.93 ± 0.52** † | 60.13 ± 0.66 | 49.97 ± 0.59 | 44.95 ± 0.56 | 53.61 ± 0.64 | 41.95 ± 0.64 |
| | No Diversity Tasks per Batch Sampler | 34.11 ± 0.55 ‡ | **66.35 ± 0.66** † | 53.77 ± 0.61 † | 57.27 ± 0.67 | 50.29 ± 0.59 | 43.91 ± 0.60 |
| | Single Batch Uniform Sampler | 26.84 ± 0.46 | 53.39 ± 0.63 | **51.14 ± 0.59** † | 54.32 ± 0.67 | 52.84 ± 0.61 | 27.16 ± 0.44 |
| | OHTM Sampler | 32.69 ± 0.52 ‡ | 64.39 ± 0.67 | 50.55 ± 0.56 † | 56.07 ± 0.67 | **58.33 ± 0.68** ‡ | 62.51 ± 0.68 |
| Meta-Dataset (Traffic Signs) | Uniform Sampler | 24.53 ± 0.39 | **48.85 ± 0.67** | **38.97 ± 0.57** | **39.59 ± 0.63** | 39.23 ± 0.58 | **52.75 ± 0.67** |
| | No Diversity Task Sampler | 25.81 ± 0.43 ‡ | 32.48 ± 0.52 | 32.84 ± 0.50 | 33.81 ± 0.51 | 35.57 ± 0.57 | 40.24 ± 0.54 |
| | No Diversity Batch Sampler | **26.16 ± 0.44** ‡ | 41.59 ± 0.59 | 38.12 ± 0.53 | 34.10 ± 0.49 | 37.67 ± 0.58 | 39.69 ± 0.55 |
| | No Diversity Tasks per Batch Sampler | 25.92 ± 0.43 ‡ | 48.73 ± 0.69 † | 38.69 ± 0.54 † | 38.86 ± 0.58 | 32.38 ± 0.47 | 35.42 ± 0.51 |
| | Single Batch Uniform Sampler | 25.62 ± 0.44 ‡ | 38.42 ± 0.54 | 39.18 ± 0.58 | 35.77 ± 0.57 | 40.87 ± 0.62 ‡ | 26.39 ± 0.45 |
| | OHTM Sampler | 24.97 ± 0.42 † | 45.73 ± 0.65 | 38.00 ± 0.51 † | 36.61 ± 0.55 | **41.56 ± 0.62** ‡ | 47.09 ± 0.63 |
| Meta-Dataset (MSCOCO) | Uniform Sampler | 23.00 ± 0.36 | 36.97 ± 0.56 | 30.82 ± 0.49 | 31.98 ± 0.55 | 33.68 ± 0.54 | 42.46 ± 0.60 |
| | No Diversity Task Sampler | 23.98 ± 0.39 † | 30.53 ± 0.51 | 30.47 ± 0.49 † | 26.37 ± 0.44 | 26.72 ± 0.46 | **42.71 ± 0.64** † |
| | No Diversity Batch Sampler | **25.95 ± 0.47** | 33.83 ± 0.53 | 29.32 ± 0.45 | 26.42 ± 0.42 | 29.25 ± 0.49 | 34.89 ± 0.54 |
| | No Diversity Tasks per Batch Sampler | 24.56 ± 0.40 † | 36.84 ± 0.55 † | 32.71 ± 0.52 † | 32.80 ± 0.57 † | 31.59 ± 0.51 | 32.88 ± 0.51 |
| | Single Batch Uniform Sampler | 23.66 ± 0.37 ‡ | 33.07 ± 0.53 | **32.47 ± 0.53** † | 33.14 ± 0.56 ‡ | **34.53 ± 0.57** † | 27.72 ± 0.44 |
| | OHTM Sampler | 23.83 ± 0.39 ‡ | **37.22 ± 0.58** | 31.02 ± 0.49 † | **33.54 ± 0.57** ‡ | 33.87 ± 0.56 † | 40.19 ± 0.60 |

Table 4: Additional Performance metrics of our models on different task samplers in the 5-way 1-shot setting.

| Dataset | Sampler | MAML | Reptile | Protonet | Matching Networks | MetaOptNet | CNAPs |
|---|---|---|---|---|---|---|---|
| Omniglot | No Diversity Tasks per Batch Sampler | 97.76 ± 0.20 | **94.55 ± 0.31** † | 97.18 ± 0.25 | 93.97 ± 0.40 | 96.80 ± 0.27 | 90.84 ± 0.47 |
| | Single Batch Uniform Sampler | 93.84 ± 0.37 | 92.60 ± 0.38 | 95.95 ± 0.31 | 92.98 ± 0.44 | 95.76 ± 0.31 | 75.86 ± 0.73 |
| | SBU unbounded | **98.06 ± 0.19** | 88.66 ± 0.55 | 97.80 ± 0.23 † | 93.82 ± 0.40 | **97.18 ± 0.26** | 89.01 ± 0.52 |
| | SBU bounded | 97.46 ± 0.21 | 87.62 ± 0.55 | **97.85 ± 0.22** † | **94.96 ± 0.37** † | 97.18 ± 0.27 | **91.24 ± 0.46** |
| MiniImagenet | No Diversity Tasks per Batch Sampler | **48.17 ± 0.62** | **41.49 ± 0.56** † | 47.73 ± 0.60 | 42.54 ± 0.53 | 50.60 ± 0.62 | **64.11 ± 0.68** † |
| | Single Batch Uniform Sampler | 41.76 ± 0.56 | 22.96 ± 0.33 | 41.35 ± 0.56 | 40.00 ± 0.54 | 39.10 ± 0.54 | 45.47 ± 0.67 |
| | SBU unbounded | 45.81 ± 0.59 | 29.82 ± 0.48 | 48.60 ± 0.60 † | 42.92 ± 0.56 | 52.24 ± 0.64 | 53.99 ± 0.67 |
| | SBU bounded | 45.65 ± 0.59 | 30.03 ± 0.49 | **48.64 ± 0.61** † | **44.21 ± 0.56** † | **52.63 ± 0.64** | 53.19 ± 0.67 |
| Tiered-Imagenet | No Diversity Tasks per Batch Sampler | **49.62 ± 0.69** | **50.77 ± 0.67** ‡ | 46.96 ± 0.67 | 43.15 ± 0.66 ‡ | 43.05 ± 0.63 | **62.81 ± 0.73** ‡ |
| | Single Batch Uniform Sampler | 41.74 ± 0.66 | 36.82 ± 0.56 | 38.22 ± 0.62 | 38.08 ± 0.61 | 36.21 ± 0.60 | 45.56 ± 0.72 |
| | SBU unbounded | 49.40 ± 0.68 | 43.68 ± 0.61 | 50.46 ± 0.68 ‡ | 43.08 ± 0.68 ‡ | **51.95 ± 0.69** ‡ | 56.23 ± 0.74 |
| | SBU bounded | 47.33 ± 0.68 | 44.51 ± 0.62 | **51.11 ± 0.69** ‡ | **43.23 ± 0.69** ‡ | 51.93 ± 0.69 ‡ | 52.77 ± 0.74 |

Table 5: Performance metric of our models on SBU oriented task samplers in the 5-way 1-shot setting.

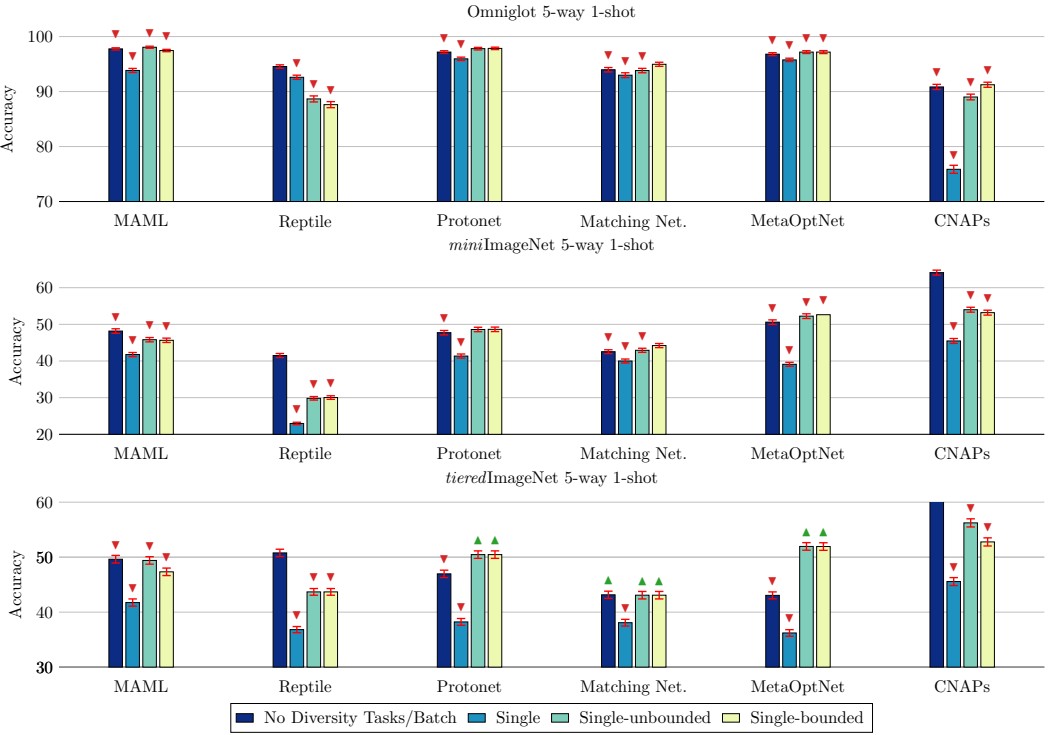

Figure 9: Average accuracy on Omniglot 5-way 1-shot, *mini*ImageNet 5-way 1-shot, & *tiered*ImageNet 5-way 1-shot with a 95% confidence interval. We denote all samplers that are worse than the Uniform Sampler and are statistically significant (with a p-value $p = 0.05$) with ▼, and those that are significantly better than the Uniform Sampler with ▲.

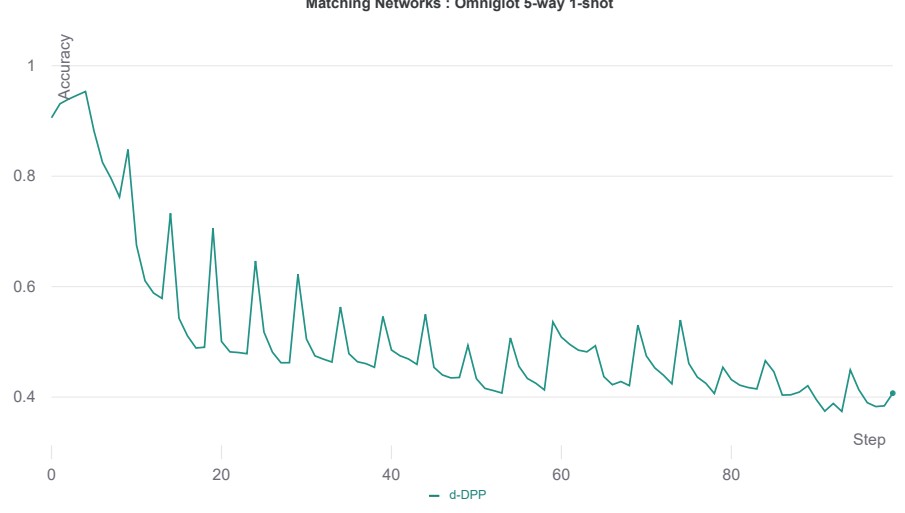

Figure 10: Convergence curve of Matching Networks model on Omniglot 5-way 1-shot.

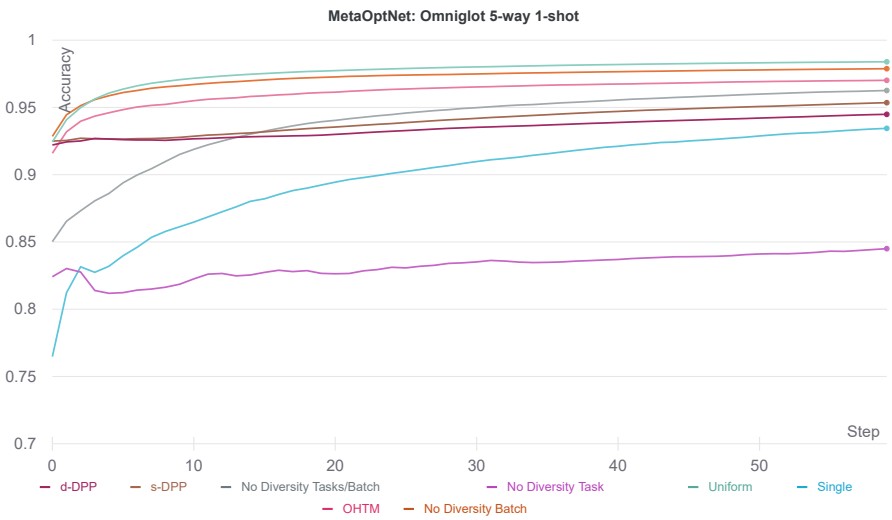

Figure 11: Convergence curve of MetaOptNet model on Omniglot 5-way 1-shot.

