# OpenReview forum: "The Effect of diversity in Meta-Learning"
_ICLR.cc/2022/Conference — ICLR 2022 Submitted_

### Official Review · Reviewer_UoR2 · 2021-11-02

**Correctness:** 3
**Technical Novelty And Significance:** 2
**Empirical Novelty And Significance:** 3
**Recommendation:** 5
**Confidence:** 3

**Main Review:**

Significance: The paper follows an existing line of work that empirically shows task diversity (in the training phase) does not help with meta-learning. This is an important step towards a better understanding of meta-learning.

Novelty: The key finding that task diversity may not be beneficial for meta-learning has been proposed and studied by Setlur et al. (2021). The paper extends the same findings to (a) different meta-learning algorithms, (b) different task samplers, and (c) different datasets. For this reason, I find the novelty of the paper limited.

There are some technical ambiguities in the paper, as discussed below.
1. The explanation of the empirical results is not insightful and convincing enough. In section 5, the authors attribute the poor (or good) performance of various diversity samplers to noise (or lack of thereof) introduced when training on multiple sub-datasets. This reason is not particularly insightful or useful to other researchers and needs more detail. For example:
(a) what noise the authors are referring to? There is no noise introduced in the problem formulation in section 3.1.
(b) how does diversity in tasks introduce noise?
(c) how does this noise affect the performance of meta-learning?
(d) can we perform ablation studies to confirm that the samplers’ poor performance is indeed caused by noise?
2. Some choice of task samplers in section 3.3 lacks motivation. For some task samplers, e.g., Online Hard Task Mining and Static DPP, the authors provide citations to various works that propose the use of these samplers. However, for the remaining samplers, the paper does not discuss why these samplers are good candidates to compare with the uniform sampler.

**Summary Of The Paper:**

The paper studies how the diversity of tasks in the training phase affects the performance of meta-learning algorithms. The paper finds negative evidence, which is consistent with Setlur et al. (2021). Compared with the existing work, the paper performs more extensive experiments with different meta-learning algorithms, different task samplers, and different datasets.

**Summary Of The Review:**

The paper provides an extensive set of empirical evidence to demonstrate that task diversity (during training) is not beneficial for meta-learning. The insights and conclusions drawn from these empirical experiments, however, are not so convincing and helpful, i.e., it does not tell other researchers how to rectify the problem or how to design better meta-learning algorithms.

---

> ### Author Response · Authors · 2021-11-20
> **Response to Reviewer UoR2**
>
> **(1) The key finding that task diversity may not be beneficial for meta-learning has been proposed and studied by Setlur et al. (2021). The paper extends the same findings to (a) different meta-learning algorithms, (b) different task samplers, and (c) different datasets. For this reason, I find the novelty of the paper limited.**
> * Setlur et al. (2021) study a specific sampler by limiting the pool of tasks. The goal of their paper has been to propose a sampler that is robust even when working with a limited pool of tasks. Furthermore, their paper empirically proves that limiting task diversity does not have adverse effects. In that aspect, we believe our paper to be fundamentally different from theirs. We attempt to study with different levels of task diversity and attempt to disprove the conventional wisdom that “task diversity is good for learning”, by considering the other end of the spectrum (increased task diversity) using samplers such as OHTM, sDPP, and dDPP - something the Setlur et al. (2021) does not explore. For this reason, we do believe our paper brings some important and novel findings.
> ---
> **(2) The explanation of the empirical results is not insightful and convincing enough. In section 5, the authors attribute the poor (or good) performance of various diversity samplers to noise (or lack of thereof) introduced when training on multiple sub-datasets. This reason is not particularly insightful or useful to other researchers and needs more detail. For example: (a) what noise the authors are referring to? There is no noise introduced in the problem formulation in section 3.1. (b) how does diversity in tasks introduce noise? (c) how does this noise affect the performance of meta-learning? (d) can we perform ablation studies to confirm that the samplers’ poor performance is indeed caused by noise?**
> * We apologize if you found any of our explanations non-insightful or convincing. Please point us to those, and we will try our best to address them. To answer some of your questions:
>    * This term has only been used when accompanied by the meta-dataset. We use this term to point out the noise the model is subject to when trained with black and white images such as Omniglot or QuickDraw, followed by RGB images such as ILSVRC or MSCOCO. In hindsight, a better term would be “variability”, and we have address this in our submission.
>    * The diversity introduces noise when the target dataset to be trained on is say ILSVRC, and we meta-train on datasets such as Omniglot or Quickdraw, which are fundamentally very different datasets.
>    * To confirm this assertion, we point you to the results obtained in Table 4: CNAPS on ILSVRC and MSCOCO with the NDT sampler. This is the perfect example of this scenario. The model has only seen one task: natural images, in this case. Hence, the model, unaffected and unconfused by the noise of Omniglot, and other datasets, is able to perform best on these two datasets. Adding multiple datasets only seems to harm performance in this scenario, so much so, that only being trained on a single task is the best sampler for the given datasets.
> ---
> **(3) Some choice of task samplers in section 3.3 lacks motivation. For some task samplers, e.g., Online Hard Task Mining and Static DPP, the authors provide citations to various works that propose the use of these samplers. However, for the remaining samplers, the paper does not discuss why these samplers are good candidates to compare with the uniform sampler.**
> * Our goal in this paper has been to study the effect of diversity in meta-learning. To this extent, all the samplers we have discussed in section 3.3 are novel and have not been proposed by any other paper. Although we do mention some references for OHTM, sDPP samplers, the respective papers propose the basic concepts we use in our sampler: such as using hard tasks or using DPP to sample tasks. Furthermore, we consider our pool of samplers to be good candidates for uniform sampler since they offer varying levels of task diversity. Samplers such as NDT, NDB, NDTB, SBU all limit task diversity, whereas samplers such as OHTM, sDPP, and dDPP promote task diversity.
> ---
> We sincerely appreciate your time and effort in reviewing our paper, as well as the constructive feedback for our work. I hope we have been able to address your concerns.

---

> > ### Comment · Reviewer_UoR2 · 2021-11-22
> > **Thank you for the response. Follow-up Comment.**
> >
> > I'd like to thank the authors for the detailed response. The authors have addressed some of the concerns that I raised in the previous review. I have some further comments on the above points.
> >
> > (1) I understand that Setlur et al. claim that "limiting task diversity does not have an adverse effect", whereas the current paper argues that "increasing task diversity does not have a beneficial effect". However, the empirical experiments and the explanations are not convincing enough for readers to believe that increasing task diversity does not help. The reasons are as follows:
> > (a) The notion of (task) diversity is not formally defined in the paper. Thus, it is somewhat heuristic, and so are the samplers. For example, the statement "samplers such as NDT, NDB, NDTB, SBU all limit task diversity, whereas samplers such as OHTM, sDPP, and dDPP promote task diversity" are not proved in a rigorous manner.
> > (b) Several works in meta-learning/few-shot learning theory show that diversity is beneficial (see some references below), albeit with a different notion of diversity (again, this is a vague statement since the notion of diversity is not defined anywhere in this paper). For this reason, I find that claiming diversity does not help is misleading.
> >
> > (2) Would the phenomenon be better referred to as "distribution shift" and not "noise". In this sense, do the authors suggest associating task diversity to distribution shift, i.e., diverse tasks have vastly different data distribution and would cause problems for learning. I think this is an interesting perspective to emphasize. But again, the authors need to be more specific (formally define diversity/distribution shift) and the claims need to be supported by evidence.
> >
> > References
> >
> > Tripuraneni, N., Jin, C., & Jordan, M. (2021, July). Provable meta-learning of linear representations. In International Conference on Machine Learning (pp. 10434-10443). PMLR.
> >
> > Du, S. S., Hu, W., Kakade, S. M., Lee, J. D., & Lei, Q. (2020). Few-shot learning via learning the representation, provably. arXiv preprint arXiv:2002.09434.

---

> > > ### Author Response · Authors · 2021-11-27
> > > **Response to Reviewer UoR2. Follow-Up Comment: 1**
> > >
> > > **(1) The notion of (task) diversity is not formally defined in the paper. Thus, it is somewhat heuristic, and so are the samplers. For example, the statement "samplers such as NDT, NDB, NDTB, SBU all limit task diversity, whereas samplers such as OHTM, sDPP, and dDPP promote task diversity" are not proved in a rigorous manner.**
> > > * Thank you very much for pointing this out to us. We understand the lack of a formal definition of diversity and that our paper has been somewhat heuristic so far. Unfortunately, we could not make this change to our rebuttal version, but we have created a new metric to define this diversity, which we will expand on in the following comment. In the following table, we summarise our results on the current samplers:
> > >
> > > |            Sampler           | Diversity |
> > > |:----------------------------:|:--------------:|
> > > |   No Diversity Task Sampler  |      0.00      |
> > > |  No Diversity Batch Sampler  |      0.00      |
> > > | Single Batch Uniform Sampler |      0.00      |
> > > | No Diversity Tasks Per Batch |  ≈ 0.00  |
> > > |        Uniform Sampler       |      1.00      |
> > > |         OHTM Sampler         |      1.69      |
> > > |         d-DPP Sampler        |      12.40     |
> > > |         s-DPP Sampler        |      12.86     |
> > >
> > >
> > > From the above table, we confirm and show rigorously that our samplers can be broadly divided into three categories:
> > > * **Low-Diversity Task Samplers:** These samplers include those with an overall diversity score less than 1. These include NDT, NDB, NDTB, and SBU Samplers.
> > > * **Standard Sampler:** This serves as our baseline and is the standard sampler used in the community - the Uniform Sampler.
> > > * **High-Diversity Task Samplers:** These samplers include those with an overall diversity score greater than 1. These include OHTM, sDPP, and dDPP Samplers.
> > > ---
> > > **(2) Several works in meta-learning/few-shot learning theory show that diversity is beneficial (see some references below), albeit with a different notion of diversity (again, this is a vague statement since the notion of diversity is not defined anywhere in this paper). For this reason, I find that claiming diversity does not help is misleading.**
> > > * This is why we believe it is essential to highlight that diversity as defined by us does not help in the case of meta-learning.
> > >    * Tripuraneni et al. (2021) showed that as the number of tasks is fixed but the number of shots increases, the generalization ability of the meta-learned regressions significantly improves. Although they define a matrix for task diversity, their finding does not show that diversity helps meta-learning, as they have increased the number of data samples.
> > >    * Du et al. (2020) show theoretically that using all the samples from the source dataset leads to a better representation, which is definitely in line with conventional wisdom. However, we and other works such as Setlur et al. (2021) empirically disprove this by showing that samplers with less diversity (by minimizing the support set pool) can achieve better performance than the standard sampler. Furthermore, we also show that increasing diversity has no positive effect either.
> > >
> > > We hope that our definition of task diversity is more robust, taking into account the number of tasks in the support set pool and the semantic diversity across tasks and batches.
> > >
> > > ---
> > >
> > > **(3) Would the phenomenon be better referred to as "distribution shift" and not "noise". In this sense, do the authors suggest associating task diversity to distribution shift, i.e., diverse tasks have vastly different data distribution and would cause problems for learning. I think this is an interesting perspective to emphasize. But again, the authors need to be more specific (formally define diversity/distribution shift) and the claims need to be supported by evidence.**
> > > * “Distribution shift” would be a perfect placeholder for “noise.” However, we do not plan on associating task diversity to distribution shift for the following reason. Distribution shift only comes into play when training on different datasets, for instance, in meta-dataset. However, when training on a single dataset, the distribution shift would always be 0 as long as we do not perform a cross-domain adaptation. For this reason, we do not associate task diversity with distribution shift. We use the term diversity to define the variety induced even when there is one dataset. For instance, although the class dogs and trucks are from the same dataset-ILSVRC, we still consider the two tasks to be diverse. To this end, the phenomenon that “distribution shift” would cause problems for learning is only a supplemental finding for the sole case of meta-dataset. A more formal definition of diversity has been formulated and followed up in the comment below.
> > > ---
> > > Again, thank you very much for your critical suggestion and criticism. It helped us formulate a metric for diversity and assert our findings more concretely.

---

> > > > ### Author Response · Authors · 2021-11-27
> > > > **Response to Reviewer UoR2. Follow-Up Comment: 2**
> > > >
> > > > Before giving a more formal definition of task diversity, we set a few more fundamental ideas required to understand our metric better.
> > > >
> > > > **Task Diversity** We define task diversity as the diversity among classes within a task. This diversity is defined as the volume of parallelepiped spanned by the embeddings of each of these classes.
> > > > $$
> > > > \mathcal{TD} \propto \left [\text{vol}(\mathcal{T})  \right ]^2
> > > > $$
> > > > where $\mathcal{T}$ is defined as $\\{ c_1,...c_N \\}$, where $N$ is the number of ways, and $c_i$ is the feature embedding of the $i^{th}$ class. These feature embeddings are pre-computed using our pre-trained Protonet model, similar to the one used in sDPP. This value is analogous to the probability of selecting a task of the following classes.
> > > >
> > > > **Task Embedding** We define task embedding as the mean embedding of class features within a task. The feature embedding of the the $i^{th}$ task is computed such that:
> > > >
> > > > $$
> > > >     \mathcal{TE} = \frac{1}{m}\sum_{i=0}^m c_i
> > > > $$
> > > > where $\mathcal{T}$ is defined as $\\{ c_1,...c_N \\}$, where $N$ is the number of ways, and $c_i$ is the feature embedding of the $i^{th}$ class. These feature embeddings are pre-computed using our pre-trained Protonet model, similar to the one used in sDPP. This value is analogous to the probability of selecting a task of the following classes.
> > > >
> > > > **Batch Diversity** We define batch diversity as the diversity among tasks within a batch. This diversity is defined as the volume of parallelepipe spanned by the task embeddings of each of these tasks within a batch.
> > > > $$
> > > >     \mathcal{BD} \propto \left [\text{vol}(\mathcal{B})  \right ]^2
> > > > $$
> > > > where $\mathcal{B}$ is defined as $\\{ t_1,...t_m \\}$, where $m$ is the number of tasks within a batch, and $t_i$ is the feature embedding of the $i^{th}$ task, $\mathcal{TE}_i$.
> > > >
> > > > **Batch Embedding** We define batch embeddings $\mathcal{BE}$ as the expected value of the embedding where the probability of each batch is proportional to the volume of the embeddings parallelepiped. This definition of probability is analogous to the ones used in traditional Determinantal Point Processes (DPPs).
> > > >
> > > > $$
> > > > \mathcal{BE} =  \mathcal{BD} \int_{i} p(t) \mathcal{TE}_i \partial t
> > > > $$
> > > >
> > > > where, $p(.)$ is the distribution derived from normalized task diversity $\mathcal{TD}$.
> > > > By definition, the batch embeddings $\mathcal{BE}$ have been defined such that the embedding is biased towards the most diverse samplers. To compute the overall diversity of our sampler, we compute the volume of the parallelepiped spanned by the batch embeddings. However, we make a slight modification, such that the length of each batch embeddings is proportional to the average batch diversity, as defined earlier.
> > > >
> > > > **Definition**
> > > >
> > > > We define the diversity of the sampler as the volume of the parallelepiped spanned by the batch embeddings.
> > > >
> > > > $$
> > > >     \mathcal{OD} \propto \left [\text{vol}(\mathcal{BE})  \right ]^2
> > > > $$
> > > > With this definition, the volume spanned will be reduced if the sampler has low diversity within a batch. Furthermore, the batch embeddings would be very similar if the model has low diversity across batches, thus reducing the practical volume spanned.
> > > >
> > > > With the following definition in place, we computed the average batch diversity across five batches, with a batch size of 8 with three different seeds. For samplers such as d-DPP and OHTM, we train on the Protonet model, since the embeddings would be similar and in the same latent space as those obtained from the other samplers which use the pre-trained Protonet model. Intuitively, $\mathcal{OD}$ measures the volume the embeddings cover in the latent space. The higher the value, the more volume has been covered in the latent space.
> > > > The average task diversity on the Omniglot dataset, scaled such that the uniform sampler has a diversity of 1, has been reported in the Table from previous comment.
> > > >
> > > >
> > > > We have added more details in our appendix section of our paper.

---

### Official Review · Reviewer_wVFn · 2021-11-02

**Correctness:** 3
**Technical Novelty And Significance:** 2
**Empirical Novelty And Significance:** 3
**Recommendation:** 3
**Confidence:** 5

**Main Review:**

In this paper, the authors investigate the effect of task diversity in the training process of meta-learning. The findings indicate that increasing task diversity during the meta-training process does not boost performance. They evaluate the performance on four few-shot image classification datasets.

Though the authors investigate several samplers, they only conduct the experiments on N-way K-shot few-shot image classification tasks. The findings do not surprise me since the meta-training process of N-way K-shot image classification essentially learns a better representation using all meta-training data samples (see the discussions in Baseline++, Meta-Baseline). Thus, using a less diverse sampler does not change the number of training samples, and thus it may not hurt the performance.

To make the findings more convincing and exciting, I suggest that the authors conduct the experiments on much more diverse tasks, e.g., regression tasks and noisy data. It would also be interesting to conduct qualitative analysis on toy tasks to understand why task diversity does not benefit the performance.

[1] Dhillon, Guneet S., Pratik Chaudhari, Avinash Ravichandran, and Stefano Soatto. "A baseline for few-shot image classification." arXiv preprint arXiv:1909.02729 (2019).

[2] Chen, Yinbo, Zhuang Liu, Huijuan Xu, Trevor Darrell, and Xiaolong Wang. "Meta-Baseline: Exploring Simple Meta-Learning for Few-Shot Learning." In Proceedings of the IEEE/CVF International Conference on Computer Vision, pp. 9062-9071. 2021.

**Summary Of The Paper:**

In this paper, the authors investigate the effect of task diversity in the training process of meta-learning. The findings indicate that increasing task diversity during the meta-training process does not boost performance. They evaluate the performance on four few-shot image classification datasets.

**Summary Of The Review:**

I would recommend rejection since the authors only investigate few-shot image classification tasks. More analysis and experiments are needed for such an analysis paper.

---

> ### Author Response · Authors · 2021-11-20
> **Response to Reviewer wVFn**
>
> **(1) The findings do not surprise me since the meta-training process of N-way K-shot image classification essentially learns a better representation using all meta-training data samples (see the discussions in Baseline++, Meta-Baseline). Thus, using a less diverse sampler does not change the number of training samples, and thus it may not hurt the performance.**
> * The model will indeed learn better representations using all the meta-training samples; for example, for metric-based methods such as Prototypical Networks and MetaOptNet, the feature extraction network shared across tasks is meta-learned based on all of the data from various tasks seen during meta-training. This is precisely this effect that was studied in Baseline++ and Meta-Baseline. However, this is different from the small training set used for adaptation only (the N-way K-shot dataset), where the amount of data is fixed here for control. Having less diversity in the tasks sampled reduces the effective amount of data the model sees during meta-training, and in this work we are specifically challenging the hypothesis that this has a significant impact on the final performance. Our findings show that increasing the data diversity (hence the effective amount of data) has no significant impact on the accuracy across multiple datasets and methods.
> ---
> **(2) To make the findings more convincing and exciting, I suggest that the authors conduct the experiments on much more diverse tasks, e.g., regression tasks and noisy data. It would also be interesting to conduct qualitative analysis on toy tasks to understand why task diversity does not benefit the performance.**
> * Thank you very much for your suggestion. We have run similar experiments with MAML and Reptile on toy examples such as Sine waves (Finn et al., 2017), Harmonic functions (Lacoste et al., 2018), and Sinusoid & lines (Finn et al., 2018). The results from our experiments can be found in Appendix A.3.1 along with Table 1, Figure 7 and Figure 8. We do not extend our DPP-oriented samplers to few-shot regression problem, as the data is continuous in nature, and computing embeddings of this continuous domain would be theoretically impossible.
> ---
> We sincerely appreciate your time and effort in reviewing our paper, as well as the constructive feedback for our work. I hope we have been able to address your concerns.

---

### Official Review · Reviewer_MHHW · 2021-11-03

**Correctness:** 3
**Technical Novelty And Significance:** 2
**Empirical Novelty And Significance:** 2
**Recommendation:** 3
**Confidence:** 4

**Details Of Ethics Concerns:**

No concerns.

**Main Review:**

In this work, the authors provide an empirical evaluation of various sampling mechanisms which can be used to create few-shot tasks during an episodic training regime that's common with meta-learning training. The list of sampling strategies include simple methods like only using a fixed set of tasks in each episode and more complicated strategies like online-hard-negative-mining. Through experiments on multiple popular few-shot learning datasets, the authors show which sampling methods do work well and which of them do not.

Strengths
--------------
* Narrative is clear and the paper is enjoyable to read. Various pictorial representations of different sampling strategies make it easy to follow.
* Experimental setup is quite exhaustive, from both the sampling mechanism point of view and also the baseline datasets chosen.

Weaknesses
------------------
* Lack of Novelty -- There is no clear novel contribution in this paper either from a theoretical or empirical perspective. The authors evaluated a series of existing task-sampling mechanisms using a bunch of existing meta-learning algorithms on few-shot benchmarks. It would have been good to see some novelty in terms of the sampling mechanism -- if uniform sampling is really the best one and that is what practitioners generally use as far as I know, then this analysis does not provide a lot of value.

* Lack of Explanation and Unfair Comparisons -- The paper does not do a good job in explaining why some of the sampling methods work better than others and in some cases, the explanations are trivial. For example, the NDT sampler performs poorly because it is only trained on a subset of data compared to the uniform sampler and therefore, is expected to provide weaker performance. However, the NDB sampler performs almost similar to uniform although it also works with limited amount of data. Why is that? What is inherently different between the subset of data selected by these two methods? One explanation I can think of is that for NDB, the initial batch contains some representations from a majority of classes in which case, its comparison to NDB is again unfair (which only contain images from a limited classes). Another similar topic is why convergence of SBU is slower than NDTB? Is it because in NDTB, each task is repeated K-times and the model has simply performed K-times more optimization steps compared to SBU? In that case, performance of SBU vs NDTB should be compared when SBU has run for k-times more epochs.

**Summary Of The Paper:**

In this work, the authors propose various task-sampling strategies for an episodic meta-learning setup and compare their performances against a standard uniform sampling. Through experiments across diverse benchmark datasets, the authors empirically show which of these methods underperform the uniform sampling and which of those perform at-par.

**Summary Of The Review:**

Although the empirical setup is well designed and the paper is easy to follow, the lack of novelty in the contribution and the apparent lack of clarity in the explanations/analysis led to my non-acceptance for this paper at this point.

---

> ### Author Response · Authors · 2021-11-20
> **Response to Reviewer MHHW**
>
> **(1) Lack of Novelty - The authors evaluated a series of existing task-sampling mechanisms using a bunch of existing meta-learning algorithms on few-shot benchmarks. It would have been good to see some novelty in terms of the sampling mechanism -- if uniform sampling is really the best one and that is what practitioners generally use as far as I know, then this analysis does not provide a lot of value.**
> * We understand the point raised by the reviewer. However, our goal in this work was to study the effect of diversity in meta-learning, not to propose a sampler that performs best. What we bring to the table is the fact that increased task diversity harms the model (which is against conventional wisdom in meta-learning), and that we can achieve similar performance to uniform sampling with only a handful of data. The first would imply that training on diverse datasets does not guarantee any boost in performance. And the creation of complicated datasets, with very diverse tasks, might not be the path to better models. The second would be a finding very useful in practical scenarios. When data is limited, and other samplers such as NDB, etc. can achieve similar performance to Uniform Sampling, it really brings to question our efficiency in using samplers such as Uniform Sampling. Although the model has seen more data, its performance is not any better. For these two reasons, we believe the analysis performed in this work does bring some practical results useful to the general audience.
> ---
> **(2) Lack of Explanation and Unfair Comparisons - The NDT sampler performs poorly because it is only trained on a subset of data compared to the uniform sampler and therefore, is expected to provide weaker performance. However, the NDB sampler performs almost similar to uniform although it also works with limited amount of data. Why is that? What is inherently different between the subset of data selected by these two methods? One explanation I can think of is that for NDB, the initial batch contains some representations from a majority of classes in which case, its comparison to NDB is again unfair (which only contain images from a limited classes).**
> * We do not believe that the explanation is as simple as that NDB has selected tasks that are the core-set of the tasks present in testing. We could argue this to be a possibility if we only tested on simple datasets such as Omniglot or MiniImagenet. However, our experiments on tieredImageNet and meta-dataset prove to us otherwise. To assure the reviewers the same, we point them towards two results:
>     * Meta-dataset results: Please refer to Figure 6, or the more detailed version in Table 4. As you can see, the datasets MSCOCO and Traffic Signs are datasets the models have never seen. It would be impossible for the tasks in training to represent any of these two datasets. Regardless, we notice that the performance of NDB is quite comparable to Uniform. Furthermore, NDT is still poor as the reviewer has rightly mentioned earlier.
>     * *tiered*ImageNet results: Please refer to Figure 6, or the more detailed version in Table 2. *tiered*ImageNet dataset is notoriously known as a harder dataset when compared to *mini*ImageNet, since the train test splits have been performed at higher level nodes. Hence, as argued by Ren et al. (2018), this split near the root of the ImageNet hierarchy results in a more challenging yet realistic regime with test classes that are less similar to the training classes. However, we observe a similar trend even in this scenario.
> ---
> **(3) Lack of Explanation and Unfair Comparisons - Another similar topic is why convergence of SBU is slower than NDTB? Is it because in NDTB, each task is repeated K-times and the model has simply performed K-times more optimization steps compared to SBU? In that case, performance of SBU vs NDTB should be compared when SBU has run for k-times more epochs.**
> * In the paper, we do mention that it might be possible that the Single Batch Uniform Sampler obtains the performance observed by the No Diversity Tasks per Batch Sampler if trained for enough epochs. However, it would be safe to comment that the convergence of the model is significantly faster in the latter. To maintain our results, we extend our experiments to two new samplers: (1) the sampler the reviewer suggests (sbu_unbounded), which would again be an unfair comparison as this sampler would have access to more data, and (2) (sbu_bounded) which addresses the above issue. We present our results and our comparison of these three samplers (including sbu) on three datasets (omniglot, *mini*ImageNet and *tiered*ImageNet) to address the point raised by the reviewer. The results from our experiments can be found in Figure 9 and Table 5 in the Appendix, along with the material at the end of Page 17.
> ---
> We sincerely appreciate your time and effort in reviewing our paper, as well as the constructive feedback for our work. I hope we have been able to address your concerns

---

### Decision · Program_Chairs · 2022-01-20

**Decision:**

Reject

**Comment:**

This paper set out to show that increasing task diversity during meta-training process does not boost performance. The reviewers mostly  agreed (only reviewer wVFn dissented) that the empirical set up of the paper was convincing, but they also felt it over-emphasized empirics over a deeper understanding of the phenomena observed. In turn, this resulted in discussions around how the experiments and the explanations didn't fully prove that increasing task diversity does not help. Overall, the discussion and the additional analysis tools provided by the authors (such as the diversity metric) will greatly improve the paper.